# ON THE CYCLE CONSISTENCY OF IMAGE-TEXT MAPPINGS

## ABSTRACT

We present an empirical study of cycle consistency in image-text mappings. We observe growing cycle consistency across a wide range of image-to-text and text-to-image models, i.e., images and text are becoming increasingly interchangeable in their representations. First, we investigate the factors driving this trend and identify that scaling language models and employing high-quality dataset recaptioning enhance cycle consistency. Next, we analyze the types of images and texts that are exchangeable, and find that cycle consistency strongly correlates with various desired properties such as reduced text hallucination, better descriptions, and improved compositionality and prompt-following in images. Lastly, we examine various sources of variance in cycle consistency demonstrating that text-to-image models are sensitive to specific prompt styles.

## 1 INTRODUCTION

How would you convey the visual look of your hometown to a friend? One approach would be to share a set of photos, showing different architectural elements and city scenes. Another would be to give a verbal description: "The roofs are made of half-cylinder terra cotta tiles, layered one on top of the other." Both approaches convey *visual* information, even though the latter is in the format of text. To wit, visual information can be communicated either by images – the preferred format of the computer vision scientist – or by language – the currency of NLP. The same is true in the other direction: information expressed via language can be visualized in an image or infographic that conveys some of the same meaning. Increasingly, multimodal models are blurring the lines between these two representational for-

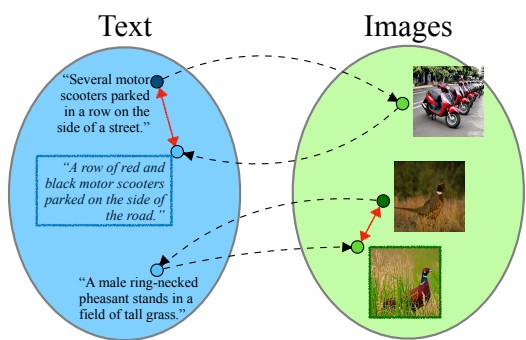

Figure 1: **Bidirectional image ↔ text mappings.** Image-text mappings are able to exchange text descriptions into images and vice versa. We analyze how close cycle reconstruction is to the original data.

mats (e.g., Liu et al., 2024b; Gemini, 2023), and there is interest in both the computer vision and NLP communities in forging links between the two modalities, so that we can apply tools from NLP to problems in vision (e.g., Surís et al., 2023), and vice versa (e.g., Hu et al., 2024).

This leads us to ask: are pixels and words fundamentally exchangeable formats, or are there limits to how effectively text can represent images, and in how well images can convey the meaning in text? We address this question by studying the degree to which images can be translated into text without losing information, and, vice versa, how faithfully can text be represented via an image. We quantify this by looking in particular at the *cycle consistency* of image-text mappings. A cycle-consistent image mapping is one in which translating from an image to a text description, and back, results in the original image. Symmetrically, a cycle-consistent text mapping translates a starting text into an image, and back into text which matches the original input. While cycle consistency is a desirable emergent property for image-text mappings, it also gives us insights about 1) what kinds of images and text lead to successful exchanges of information and 2) the models which generated these images and text. Furthermore, recent models have begun to incorporate cycle consistency at training time Betker et al. (2023); Esser et al. (2024); Sharifzadeh et al. (2024); Li et al. (2024b).

| Input Image | Reconstructed Image | Input Image | Reconstructed Image |
|---|---|---|---|

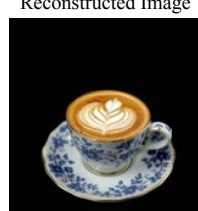 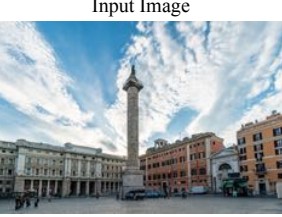 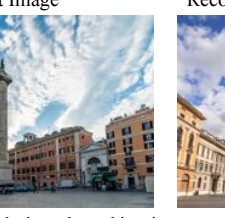

[**0.181**] The image depicts a beautifully crafted **latte** art coffee cup placed on a saucer. The cup and saucer are adorned with intricate **blue floral** patterns, giving them a classic and elegant appearance. The latte art on the coffee is a **leaf design**, crafted with precision, featuring a swirl of white foam on top of a rich, **creamy brown coffee**. The background is **dark**…

[**0.232**] The image depicts a large, historic square with a prominent **obelisk** in the center. The obelisk is tall and cylindrical, made of **stone**, and topped with a **statue**. The sky above is partly cloudy with patches of **blue**, suggesting a partly **sunny** day. Surrounding the obelisk are several **buildings** with distinct architectural styles. On the left side…

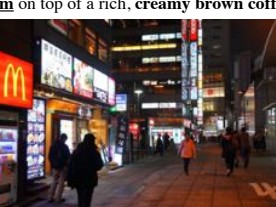 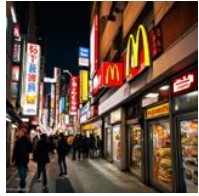 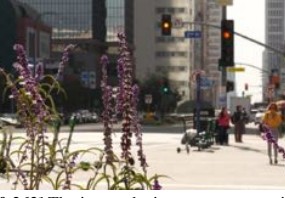 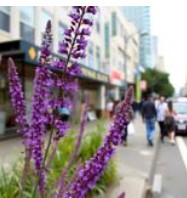

[**0.208**] The image depicts a bustling **urban street scene** at **night**, likely in a busy commercial area. The street is lined with various **shops and restaurants**, each adorned with **bright**, **colorful** signage in both **English and Japanese**. The most prominent **sign** in the image is the iconic red and **yellow McDonald's** logo, indicating the presence of a **McDonald's** restaurant. Adjacent to it…

[**0.262**] The image depicts a street scene in an **urban** environment, likely in a **city**. The foreground is dominated by a **vibrant purple flowering plant** with long, **slender** stems and clusters of **small, purple flowers**. The plant is in sharp focus, with its vibrant **purple** blossoms standing out against the **blurred background**. The background features a **busy street scene** with several **pedestrians** and **vehicles**…

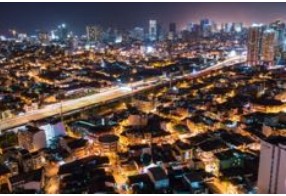 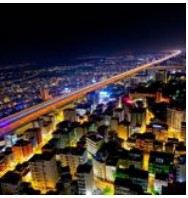 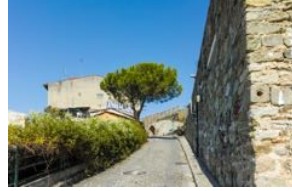 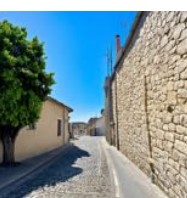

[**0.282**] The image depicts a **vibrant cityscape** at **night**, showcasing a bustling **urban** environment illuminated by **numerous lights**. The foreground features a **dense cluster of buildings**, including residential and commercial structures, with **varying heights and architectural styles**. The streets are filled with **bright lights**, indicating active businesses and possibly nightlife. A **prominent highway runs horizontally** across the middle of the image, with **streaks of light**…

[**0.347**] The image depicts a narrow, **cobblestone** street leading towards a **stone wall** on the **right** side and a building on the **left** side. The street is flanked by a large, **lush green tree** on the **left**, which stands out prominently against the **clear blue sky**. The building on the **left** appears to be a residential or commercial structure with a **flat roof** and a few **antennas** on top…

Figure 2: **Examples of image cycle consistency** using different image-to-text, text-to-image combinations. No models have been trained jointly, but still produce cycle consistent outputs. Each image pair shows the input real image, the output generated image. The text description used to generate the output image is shown below the image pair along with DreamSim(↓) reconstruction in brackets.

| Generated Image | Input and Reconstructed Text | Generated Image | Input and Reconstructed Text |
|---|---|---|---|

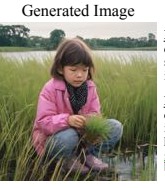 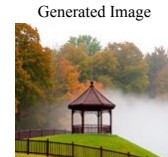

**Input text:** A young girl in a pink jacket and polka dot scarf crouches in a field of tall grass, holding a small **plant**, surrounded by lush greenery and possibly a small **body of water** in the background.

**Reconstructed text** [**0.850**]: The image depicts a young girl crouching in a field of tall green grass near a **body of water**. She is wearing a pink jacket, blue jeans, and purple shoes. The girl has short, dark hair and is holding a small **plant** or clump of grass in her hands, examining it closely. She is also wearing a **black** scarf with white polka dots around her neck

**Input text:** A gazebo sits on a grassy hill surrounded by a brown fence, with **trees** in the background displaying various colors of fall foliage. The sky is overcast, and fog rolls in, obscuring the distance.

**Reconstructed text** [**0.829**]: The image captures a serene scene of a gazebo perched on a **hill**, enveloped by a foggy atmosphere. The gazebo, with its brown roof and **black railing**, stands out against the vibrant colors of the surrounding **trees**. These **trees**, adorned in hues of orange, yellow, and green, hint at the season being autumn. The **hill** on which the gazebo is located…

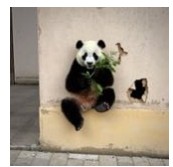 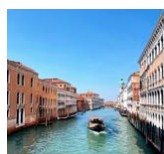

**Input text:** A **panda bear** sits on a **damaged wall**, holding a **plant**, with a hole nearby. The panda's face and paw are visible, and the image includes a sidewalk and a lower wall section.

**Reconstructed text** [**0.840**]: The image depicts a **panda bear** sitting on a wall. The **panda bear** is holding a **green plant** in its paws, and it appears to be sitting on a crack in the wall. The wall is painted in a light color, possibly beige or light brown, and there is a noticeable hole in the wall where the panda is perched…

**Input text:** A scenic view of **Venice, Italy**, with buildings along the **canal** and a **small boat** sailing on the water. The buildings have an **architectural** style common in **Venice**, and there is a large teal-colored dome on top of one of the buildings. The sky is a rich blue with no clouds.

**Reconstructed text** [**0.858**]: The image depicts a picturesque scene of a **canal** in **Venice, Italy**. The **canal** is flanked on both sides by historic buildings, showcasing the city's iconic **architecture**. The water is a serene, turquoise color, reflecting the **clear blue sky** above. A **small boat**, possibly a gondola, is seen moving along the canal, leaving a gentle wake behind it

Figure 3: **Examples of text cycle consistency** from different model combinations. Models still exhibit a degree of cycle consistency despite being trained independently. Each example shows the generated image on the left for the input text on the right, followed by the reconstructed text below. SBERT similarity (↑) between the input and output text is reported in brackets. Highlighted phrases in the descriptions are for better comparison.

We make the following findings. First, we experiment with combining off-the-shelf text-to-image and image-to-text models to create both image and text cycles. We find that current models are fairly cycle-consistent semantically, but still distant from pixel to pixel (as seen in Figures 2 and 3). Furthermore, we observe an increasing correlation between cycle consistency and model performance. We analyze several key advancements contributing to this trend: language model scale,

higher resolutions, and training data with densely captioned images. Secondly, we observe that cycle-consistent captions are descriptive, exhibit reduced object hallucination and omission, and dense in length. Cycle-consistent images demonstrate better prompt-following and improved compositionality across different categories. Because image-text mappings are not one-to-one, we compare sources of variance in these mappings, including forms of sampling and prompt style choices, and their effect on cycle consistency.

## 2 PRELIMINARIES

We examine to what degree current models display cycle-consistent properties. Similar to how autoencoders calculate error between original inputs and decoded outputs to evaluate performance, we measure both how well are images reconstructed through text and how well are text descriptions preserved by images. We use $I$ to denote a set of real images, and $T$ to denote a set of text descriptions. Given an image-to-text model $F$ and a text-to-image model $G$, they exhibit *cycle consistency* if $G(F(i)) \approx i$ for all $i \in I$ and $F(G(t)) \approx t$ for all $t \in T$. We measure *image cycle consistency* and *text cycle consistency* by computing the following reconstruction losses respectively:

$$\mathcal{L}_{\text{img}} = \mathbb{E}_{i \in I}[d_{\text{img}}(G(F(i)), i)], \tag{1}$$

$$\mathcal{L}_{\text{text}} = \mathbb{E}_{t \in T}[d_{\text{text}}(F(G(t)), t)]. \tag{2}$$

We measure the distance between image $i$ and reconstructed image $G(F(i))$ with the DreamSim (Fu et al., 2023b) image distance metric $d_{\text{img}}$. We find that DreamSim cycle consistency best correlates with text descriptiveness and aligns with human perception of image distance. Similarly, we use SBERT (Reimers & Gurevych, 2019) to measure similarity between input and reconstructed text. See Appendix C.1 for ablations on measuring cycle consistency.

## 3 WHAT FACTORS ARE DRIVING CYCLE CONSISTENCY?

In this section, we analyze the driving factors for cycle consistency. We evaluate image and text cycle consistency for 13 image-to-text models and 5 text-to-image models (i.e., 130 cycle-consistent mappings). These models were trained with varying datasets, architecture, and scale, and were selected based on public availability and disclosure of details. See Appendix A for a complete list of models and summary of differences.

We use Densely Captioned Images (DCI) dataset (Urbanek et al., 2024), which features high-resolution images annotated with dense captions, compared to other datasets (e.g., 480×640 pixels, 13.54 tokens for MSCOCO (Lin et al., 2014)). Due to limited prompt length of text-to-image models, we use sDCI which summarizes DCI captions to fit 77 tokens (1500×2250 pixels, 49.21 tokens). We sample 1K examples from the train split. We report the average cycle consistency for each text-to-image model, computed across 13 image-to-text models and 3 random seeds. We follow the same procedure with image-to-text models. Image and text cycle consistency calculations for all possible model combinations are shown in Figures 14, 15. Figure 16 provides a baseline comparison.

### 3.1 CYCLE CONSISTENCY IMPROVES WITH LLM SCALE

An image-to-text model consists of a vision encoder, a projector, and a large language model (LLM). Scaling the vision transformer (ViT) for the vision encoder is reported to enhance performance (Li et al., 2023b), yet a simple MLP projection remains the dominant approach (Liu et al., 2023b;a; OpenGVLab, 2024; Li et al., 2024a). As no model offers open-sourced weights with varying vision encoder scales while keeping other parameters fixed, we focus our analysis on ablating the LLM size. Figure 4 demonstrates that scaling the LLM enhances both image and text cycle consistency across all image-to-text model families. Figure 5 highlights the effect of LLM size on image cycle consistency, comparing the InternVL2 model family trained on the same architecture and dataset but with varying LLM scales. We observe that scaling the LLM improves caption descriptiveness, e.g., InternVL2-40B is the only model capable of accurately describing both the color and the presence of a corner turret. In contrast, models with smaller LLMs fail to capture such fine-grained details, leading to reduced image cycle consistency.

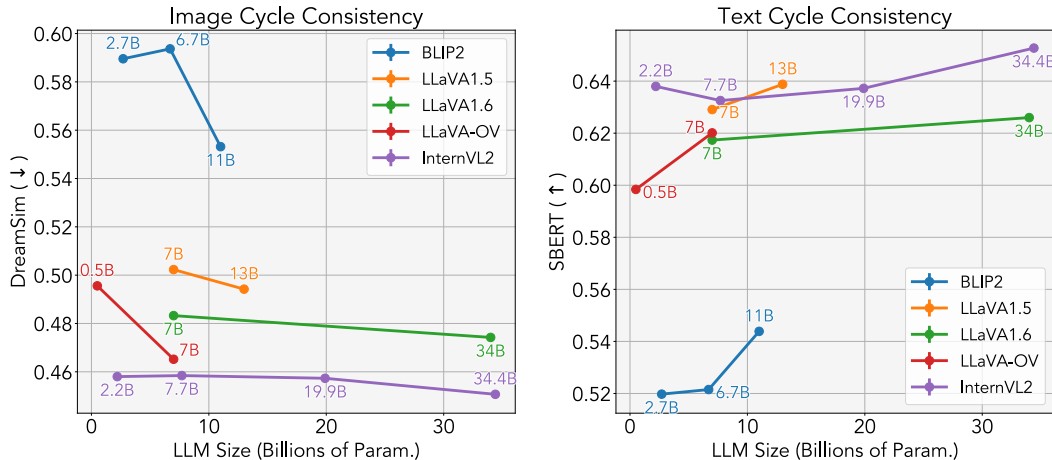

Figure 4: **Scaling the language model improves cycle consistency.** Across all image-to-text model families, both image and text cycle consistency consistently improve with the scaling of large language models (LLMs).

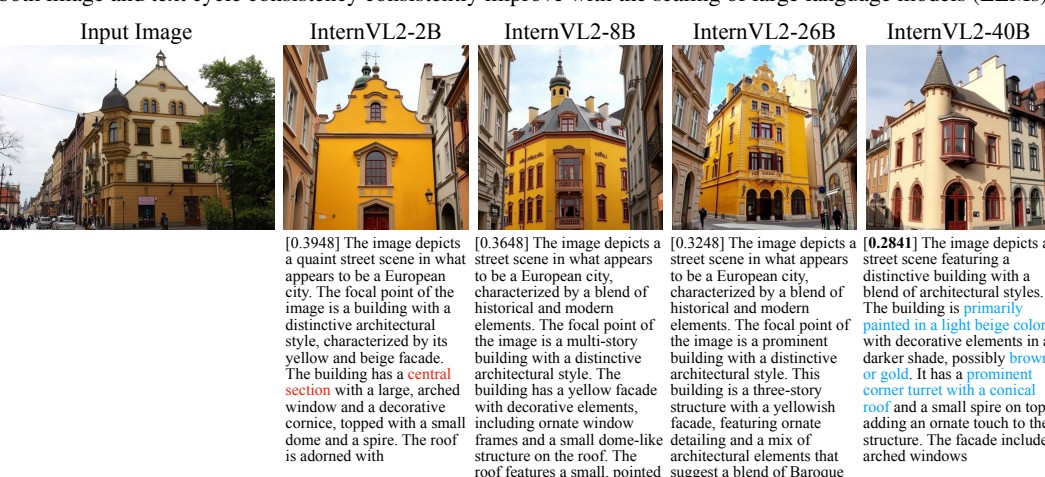

Figure 5: **How LLM scale affects image cycle consistency.** Left to right: input image followed by four image reconstructions using synthetic captions from image-to-text models with different LLM size. Each caption is preceded by the respective DreamSim(↓) reconstruction score in brackets. Despite being trained on the same dataset and architecture, scaling the LLM improves caption descriptiveness. For instance, only InternVL2-40B successfully captures both the color and the presence of a corner turret, whereas models with smaller LLMs lack fine-grained detail, resulting in poorer image cycle consistency.

## 3.2 RE-CAPTIONED DATASET QUALITY

Current image-to-text models are predominantly trained on re-captioned datasets where real images are annotated with *detailed descriptions* generated by large language models (e.g., GPT-4) or vision-language models (e.g., GPT-4V). Similarly, recent works demonstrate that training text-to-image models with descriptive captions generated by high-quality captioning models significantly enhances their prompt-following ability (Betker et al., 2023; Esser et al., 2024). As most text-to-image models do not disclose information on training data, our analysis primarily focuses on image-to-text models and their re-captioned datasets. Ideally, the analysis would involve the same model trained with and without re-captioned datasets, or with datasets of varying quality; however, such open-sourced weights are not available. Consequently, we compare different models of similar sizes. To limit the number of uncontrollable variables, we compare models with similar number of parameters. Note that other factors, such as architecture, pre-trained backbones, and model training still differ between the models.

Table 1 demonstrates that the quality of the re-captioned dataset (e.g., dataset re-captioned by GPT-4V, LLaVA1.6-34B) aligns with improved image cycle consistency. Models trained on such datasets often exhibit better consistency than those trained on larger datasets annotated by less-performant

| Model | Re-captioned Dataset | | Cycle Consistency | |
| | Size | Re-captioning Model | Image (↓) | Text (↑) |
|---|---|---|---|---|
| BLIP2-6.7B | 244M | BLIP | 0.5936 | 0.5215 |
| LLaVA1.5-7B | 23K | GPT-4* | 0.5022 | **0.6290** |
| LLaVA1.6-7B | 112K | GPT-4V | 0.4833 | 0.6173 |
| LLaVA-OV-7B | 3.5M | LLaVA1.6-34B | **0.4742** | 0.6259 |

Table 1: **Re-captioned dataset quality and cycle consistency.** The quality of the re-captioned dataset (i.e., generated by a high-performing model) aligns with improved image cycle consistency. Models trained on such datasets often exhibit better consistency than those trained on larger datasets annotated by less-performant models. In contrast, text cycle consistency shows little difference between the LLaVA models due to limited descriptiveness of the input text (sDCI), resulting in diminishing improvements for longer, more detailed captions, such as those reconstructed by LLaVA1.6 and LLaVA-OV. Image cycle consistency is measured by DreamSim (lower is better), and text cycle consistency by SBERT (higher is better).

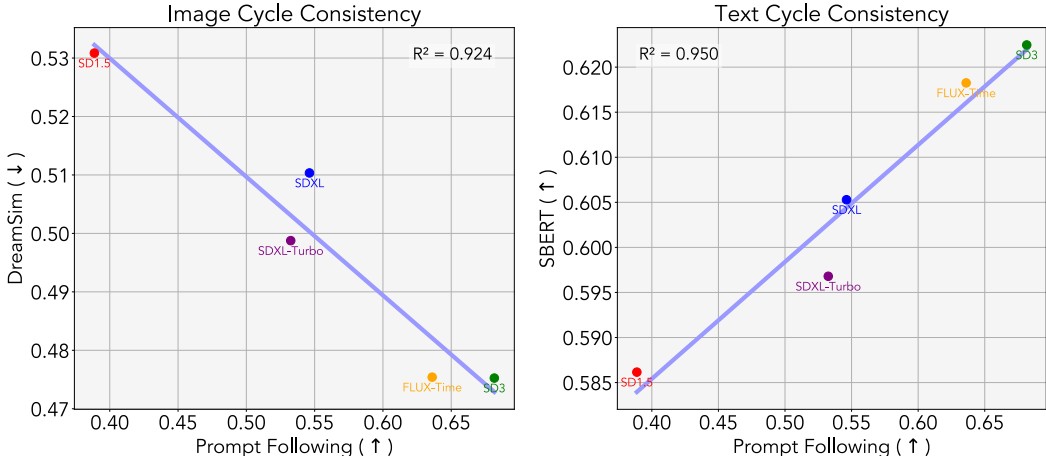

Figure 6: **Cycle consistency correlates with better prompt-following images.** We measure cycle consistency (averaged across all image-to-text models) as a function of prompt-following quality in generated images. We observe a strong correlations for both kinds of cycle consistency and prompt-following quality in images.

models (e.g., BLIP). Our findings align with existing literature (Li et al., 2024a) that highlights "quality over quantity" for training multimodal models. On the other hand, text cycle consistency shows little difference between the LLaVA models, as the input text from sDCI often lacks fine-grained detail (evidenced in Figure 16) compared to longer and more descriptive synthetic captions, such as those produced by LLaVA1.6 and LLaVA-OV. We believe higher-quality human annotations and text-to-image models with longer context would enhance the analysis of text cycle consistency. Note that for LLaVA1.5, GPT-4 (i.e., language model) is prompted with captions and bounding boxes to generate detailed captions. We exclude InternVL2 from this analysis as information regarding its pre-training dataset is not disclosed.

## 4 PROPERTIES OF CYCLE-CONSISTENT IMAGES AND TEXTS

In this section, we explore the quality of cycle-consistent texts and images with respect to various properties. We find more descriptive and less-hallucinated captions and better prompt-following images generally align with higher cycle consistency.

### 4.1 COMPOSITIONALITY AND PROMPT-FOLLOWING IN IMAGES

We investigate how cycle consistency varies with compositionality and prompt-following in text-to-image generation. We measure these qualities on two benchmarks: T2I-Compbench (Huang et al., 2023), which focuses on image compositionality, and Drawbench which includes a variety of categories for general text-to-image synthesis (Saharia et al., 2022).

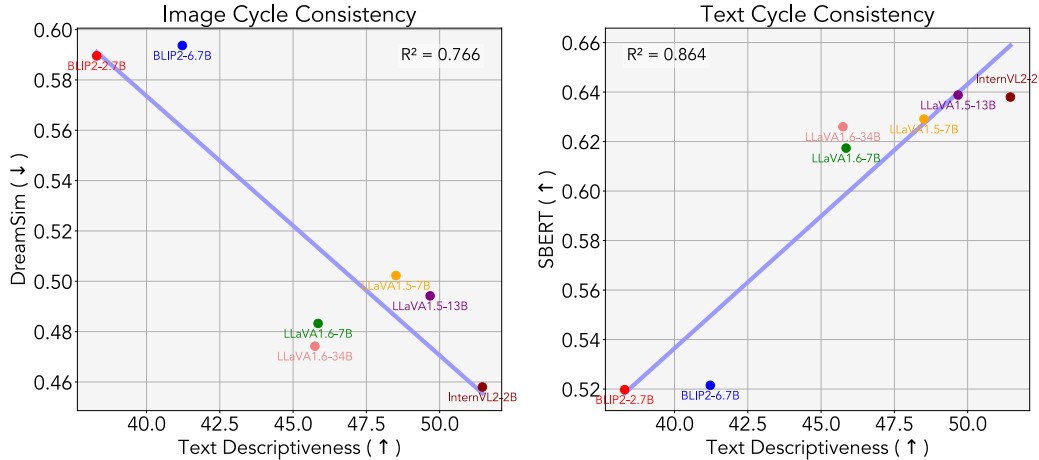

Figure 7: **Cycle consistency strongly correlates with descriptiveness in text.** While both cycles exhibit a strong correlation, we observe a better alignment between text cycle consistency and descriptiveness in the generated captions.

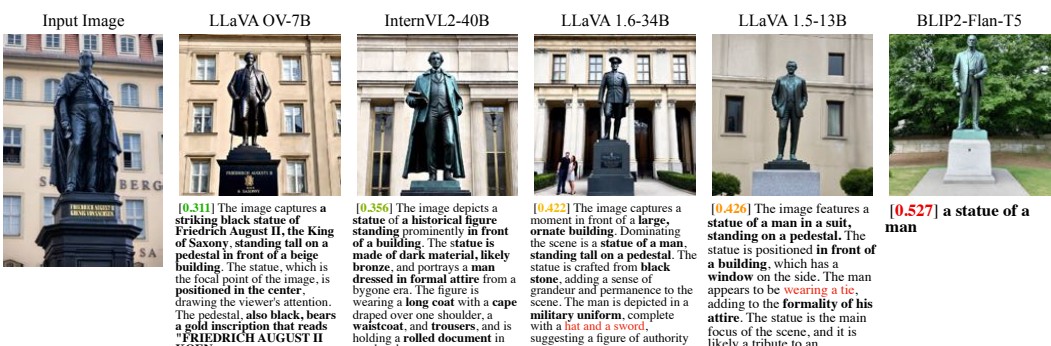

Figure 8: **Text descriptiveness and cycle consistency**. From left to right: Original photograph, and generated image reconstructions made by different captions under each image with the DreamSim ($\downarrow$) reconstruction in brackets. As captions include more specific and correct visual details, the reconstruction quality increases. All images are generated with SD3 from the same random seed.

For T2I-Compbench, we evaluate on color, shape, texture, and spatial fine-grained categories. Similarly to Section 3 we calculate cycle consistency for text-to-image models by averaging across all 13 image-to-text models. Figure 6 plots image and text cycle consistency against text-to-image performance for all 5 text-to-image models. Cycle consistency highly correlates with text-to-image generation quality. Intuitively, images which are more faithful to visual details represented in text preserve more information and therefore facilitate better image and text reconstructions. Ablations for measuring cycle consistency are discussed in Appendix C.1.

## 4.2 TEXT DESCRIPTIVENESS

Similarly to Section 4.1, we analyze the relationship between the descriptiveness of captions and cycle consistency. Typically, image captions are evaluated using the CIDEr score on the COCO Karpathy split (Karpathy & Fei-Fei, 2015). This dataset covers a limited distribution of images annotated with short captions which poses a challenge in evaluating modern image-to-text models which generate long, descriptive captions. Inspired by recent text-to-image benchmarks (Huang et al., 2023; Saharia et al., 2022), we instead conduct visual question answering without the image component, i.e., "VQA without V". Given a VQA dataset $\{v, q, a\}_{i=1}^{N}$, an image-to-text model $F$, we first generate synthetic text $F(v)$ for images in the VQA dataset. Then we prompt a large language model (LLM) to *answer the question based on the generated caption*. This allows us to measure whether synthetic text accurately describes fine-grained details of the image, as the LLM must answer a diverse range of questions based solely on the description. We use Meta-Llama-3.1-8B-Instruct (Dubey et al., 2024) as the LLM evaluator in all experiments. Exact prompts are detailed in Appendix A.3.

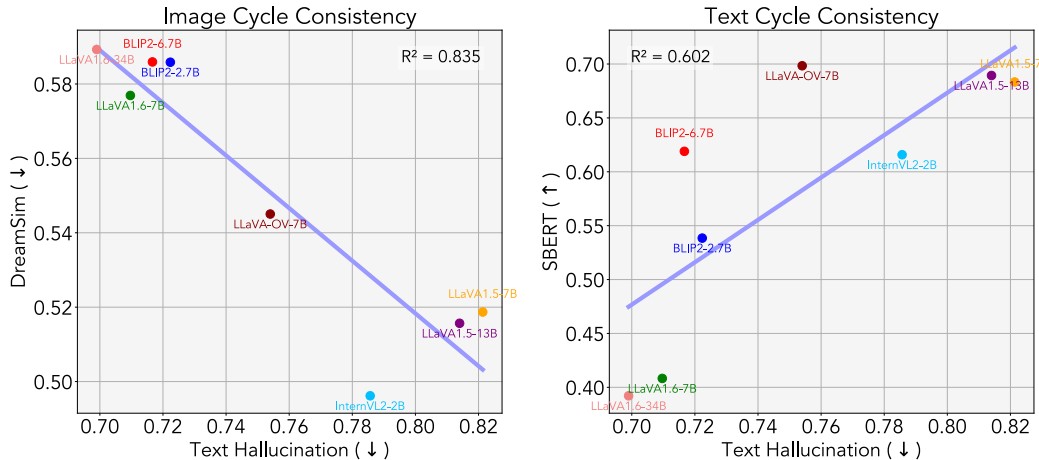

Figure 10: **Cycle consistency strongly correlates with reduced hallucination in text**. We observe that both cycles strongly correlates with reduced hallucination in text, with image cycle consistency being more prominent.

To assess how informative synthetic text is across fine-grained attributes (e.g., counting, position), we perform "VQA without V" on three VQA benchmarks: SEEDBench (Li et al., 2023a), MME (Fu et al., 2023a), and MMStar (Chen et al., 2024a). As we evaluate how well synthetic text describes images, we focus on questions from the "perception" categories across all datasets. Cycle consistency is calculated as in Section 3.

Figure 7 compares image-to-text model captioning performance scores against image and text cycle consistency scores (averaged across all text-to-image models). This demonstrates a strong correlation between both image and text cycle consistency and captioning performance. Generally, more informative captions lead to better image reconstructions through text, and also better recovery of input text details such as in Figure 8. For a discussion of ablations measuring cycle consistency see Appendix C.1.

**Sometimes less is more.** Although generally descriptive captions exhibit better cycle consistency, we observe examples of high cycle consistency using short, undescriptive captions. Such instances are not uncommon, and mainly occur when the input image displays a very typical scene. In Figure 9 (Top Row), images of large golden statues are often taken by people at ground level looking up at the statue. Perhaps such images are "common" enough that a few keywords are sufficient to fully describe the scene. See Appendix Figure 22 for more examples.

Input Image     Reconstructed Image

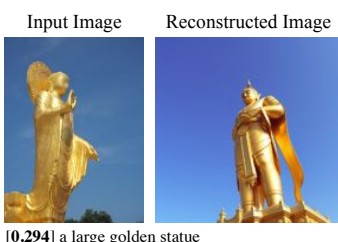

[**0.294**] a large golden statue

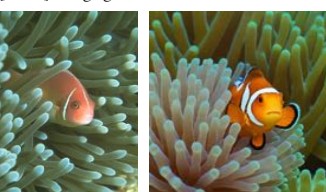

[**0.274**] Anemone, Clownfish

Figure 9: **Short text captions** can yield high image cycle consistency despite lack of detail. Strong bias between certain images and text captions leads to easy reconstructions.

### 4.3 OBJECT HALLUCINATION IN TEXT

To investigate object hallucination in the generated text, we utilize the POPE benchmark (Li et al., 2023c), using their annotated COCO dataset. POPE constructs a set of triples consisting of an image, multiple questions and their answers $\langle x, \{q(o_i), a_i\}_{i=1}^{l} \rangle$, where $x$ is the image, $q(o_i)$ is a question probing object $o_i$ based on a template *"Is there a/an <object> in the image?"*, $o_i$ is the $i$-th object to be probed, and $a_i$ is the answer to the question ("Yes" or "No"), and $l$ denotes the number of questions per image. While POPE focuses on evaluating hallucination at the *model* level, our goal is to measure hallucination in the *generated text*.

To achieve this, we perform the VQA without V analysis on POPE. Specifically, for a given LLM $M$, we evaluate $M(f(x), q(o_i)) = a_i$, which measures how many questions can be answered based on the generated caption. This contrasts with evaluating $f(x, q(o_i)) = a_i$, which measures how accurately an image-to-text model $f$ can answer questions directly. The experimental setup is identical to that described in Section 4.2. Figure 10, 11 demonstrates that cycle consistency strongly

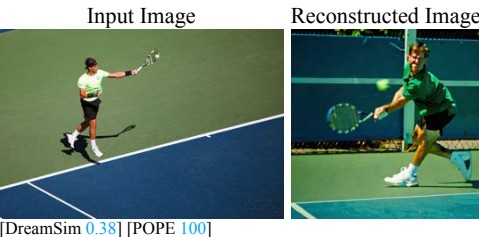

Input Image     Reconstructed Image      Input Image     Reconstructed Image

[DreamSim 0.38] [POPE 100]
a man in a green shirt and black shorts is playing tennis on a blue court. he is holding a tennis racket and appears to be in the middle of a swing.

[DreamSim 0.41] [POPE 83.3]
three wine glasses are filled with red wine and placed on a table. the glasses are arranged in a row, with one glass on the left, another in the middle, and the third on the right. the table setting also includes a cup and a bowl, creating a cozy and inviting atmosphere for a meal or gathering.

Figure 11: Reduced object hallucination (POPE ↑) correlates with better cycle consistency (DreamSim ↓).

correlates with texts with reduced hallucination. Notably, we observe that *image* cycle consistency shows a stronger correlation with reduced hallucination.

## 4.4 TEXT DENSITY

Given that more informative captions correlate with more cycle consistent images as seen in Section 4.2, we now study how densely information should be packed into a caption - i.e., to get the best image reconstruction, what is the ideal caption length? To properly control both the level of detail and length of captions, we use summarized captions for images in the Densely Captioned Images dataset (Urbanek et al., 2024) created by Huh et al. (2024). In this dataset, each image is accompanied by captions of different lengths: 5, 10, 20, 30, and 50 word summaries of the original fully detailed DCI caption. For each captions of each summary length, we use text-to-image models to generate images over 10 different random seeds and then report the average DreamSim score between the generated images and the original image described by the captions. Figure 12 plots the relationship between caption density and image reconstruction. Similarly to Section 4.2, we find that increasing the amount of granularity of captions improves reconstruction error for all text-to-image models, although with high variance. Furthermore, for models `SD1.5`, `SDXL`, and `SDXL-Turbo` image reconstruction sees little benefit beyond 30 tokens, whereas `SD3` and `FLUX-Time` continue to show improvement. Figure 13 provides examples of summary captions and their corresponding synthetic images.

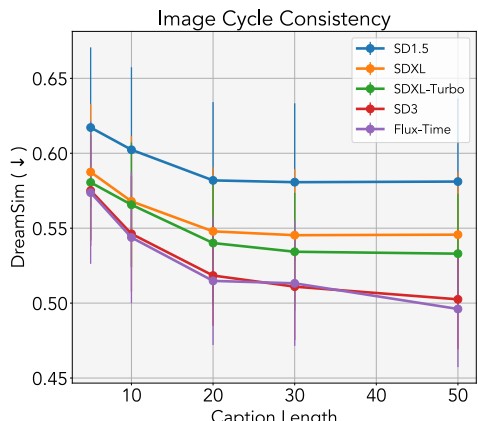

Figure 12: **Image cycle consistency vs. caption density.** We measure cycle consistency across captions for the same image summarized into varying lengths. Reconstruction scores increase with more descriptive captions, with reduced benefit after about 30 words. Error bars show standard deviation across 10 different random seeds.

## 5 VARIANCE IN CYCLE CONSISTENCY

In the previous sections, we mainly observe cycle consistency by greedy sampling from image-to-text models and averaging over three random seeds for text-to-image models. In this section, we observe how stochasticity can affect cycle consistency. Text-to-image and image-to-text models can exhibit stochastic behavior due to factors such as random seed initialization, temperature sampling, and differences in prompt wording. While random seed and temperature sampling are only relevant to one mapping direction, prompt style applies to both. Different prompt styles for text-to-image generation come from changing the text while maintaining its meaning. One such example is editing word choice and syntax. For image-to-text models, input prompts can be used to query the model and request different kinds of text descriptions. These sources of variability can lead to different results given identical inputs. In this section, we analyze the extent to which each factor causes variance in cycle consistency.

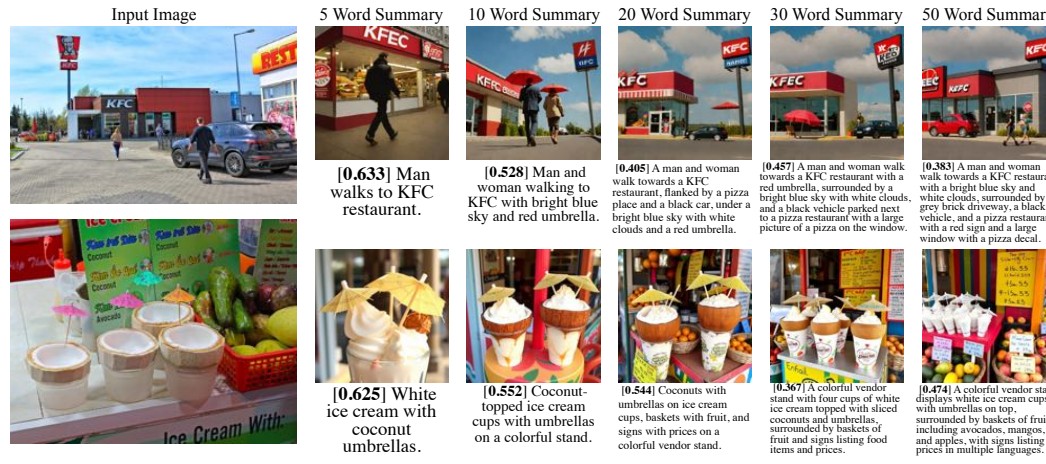

Figure 13: Effect of caption density on image cycle consistency. From the left to right: Original photograph, and synthetic reconstructions made by different summarized versions of the same caption. Under each synthetic image is the DreamSim reconstruction followed by the summary caption. Generally as the captions become longer and more descriptive, they better match the original image.

| Source of Variance | Image Cycle Consistency | Text Cycle Consistency |
|---|---|---|
| **Text-to-image Models** | | |
| Random seed | 0.0415 | 0.0634 |
| Caption style | **0.0709** | **0.0719** |
| **Image-to-text Models** | | |
| Temperature sampling | **0.0642** | **0.0588** |
| Prompt style | 0.0371 | 0.0427 |

Table 2: **Sources of variance in cycle consistency.** We compare how random seed, prompt style, and temperature sampling affect variance in cycle consistency. For each source of variance we report the average standard deviation using DreamSim and SBERT for image and text cycle consistency respectively.

For each factor, we generate $N = 10$ variations and calculate the average standard deviation using DreamSim for image cycle consistency and SBERT for text cycle consistency. For temperature sampling, we set the temperature to 0.7. For text-to-image models, we modify prompt style by using Meta-Llama-3.1-8B-Instruct (Dubey et al., 2024) to rewrite a given prompt while maintaining its original meaning and number of words. The choice of prompts for the models is included in the Appendix A.1. We use the DCI dataset and sample 100 examples with $N = 10$ variations each across 60 model combinations, resulting in variance calculations over 6,000 examples. Note that we excluded BLIP2 models from measuring prompt style variance as they are not instruction-tuned and often produce captions of less than 3 words, making it challenging to change the style without changing its meaning.

Table 2 demonstrates that image-to-text models exhibit higher variance due to temperature sampling but remain relatively robust to changes in prompt style. In contrast, text-to-image models are significantly more sensitive to prompt style than random seed sampling.

## 6 RELATED WORK

Large multimodal models are rapidly improving, particularly in vision and language. Image-to-text models are capable of producing comprehensive image descriptions (Gemini, 2023; OpenAI) by scaling the language model (Liu et al., 2023b;a) and training on semantically-rich synthetic captions (Li et al., 2022; 2023b; Sharifzadeh et al., 2024; Liu et al., 2023b;a).

Concurrently, text-to-image models can generate images that follow a wide range of prompts (Podell et al., 2023; Sauer et al., 2023; Esser et al., 2024; BlackForestLabs). Recent works (Betker et al., 2023; Brooks et al., 2024) further enhance the prompt-following ability by learning a descriptive

image captioner and generating pseudo image-text pairs. The models trained on this dataset are capable of generating images faithful to long, descriptive captions.

While several works implicitly encourage image-text cycle consistency during sampling, recent work (Li et al., 2024b) explicitly enforces cycle consistency by leveraging unpaired image and text data, ensuring alignment between the original samples and their cycle-generated counterparts. Despite large image-text models becoming increasingly cycle-consistent, this property has been surprisingly little studied. In this work, we provide an in-depth analysis of cycle-consistent properties across a wide range of off-the-shelf image-to-text and text-to-image models.

# 7 CONCLUSION

This paper studies cycle consistency in current image-text mappings. We find that existing image-to-text and text-to-image models achieve a certain level of cycle consistency, even without explicit training for it. We observe that image-to-text models with larger LLMs trained on high-quality re-captioned data are associated with higher cycle consistency. Moreover, we show that cycle consistency improves with the quality of synthetic text and image generations. Generated images that follow the compositionality and details provided by the input text model tend to be more cycle-consistent, and similarly for more detailed, informative, and accurate captions. Lastly, we highlight stochastic factors that may affect cycle consistency, and find that prompt style for text-to-image models contributes the most variance to cycle consistency.

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

# A    MODEL DETAILS

We use 13 different image-to-text models and 5 text-to-image models to study cycle consistency. Models are chosen based on public availability and disclosure of details such as architecture, scale, training dataset and method etc. We use the following models for image-to-text mappings: BLIP-2.7B, BLIP-6.7B, BLIP-2-Flan T5-XXL Li et al. (2023b), LLaVA 1.5-7B, LLaVA 1.5-13B Liu et al. (2023a), LLaVA OneVision-Qwen2-0.5B, LLaVA OneVision-Qwen2-7B Li et al. (2024a), LLaVA 1.6 Mistral-7B, LLaVA 1.6-34B Liu et al. (2024a), InternVL2-2B, InternVL2-8B, InternVL2-26B, and InternVL2-40B Chen et al. (2023; 2024b). Table 5 provides a summary of model differences in terms of scale and architecture.

We use the following models for text-image mappings: Stable Diffusion 1.5 (Rombach et al., 2022), Stable Diffusion XL (Podell et al., 2023), Stable Diffusion XL Turbo (Sauer et al., 2023), Stable Diffusion 3 (Esser et al., 2024), and FLUX (Timestep-distilled) (BlackForestLabs). Tables 6 provides a summary of model differences.

## A.1    HYPERPARAMETERS FOR IMAGE-TO-TEXT MODELS

To ensure that all image-to-text models can produce image descriptions to the best of their ability, we use the prompt recommended by the model distributor, as shown in Table 3. We use greedy search for all experiments (except for temperature sampling in Section 5), and 77 maximum tokens, i.e., maximum prompt length supported by text-to-image models.

| Model | Prompt |
|---|---|
| BLIP2 | "this is a picture of" |
| LLaVA1.5 | "Write a detailed description of the given image." |
| LLaVA1.6 | "Write a detailed description of the given image." |
| LLaVA-OV | "Write a detailed description of the given image." |
| InternVL2 | "Please describe the image in detail." |

Table 3: Prompts used for generating image descriptions for image-to-text models.

## A.2    HYPERPARAMETERS FOR TEXT-TO-IMAGE MODELS

To ensure that all text-to-image models can produce outputs to the best of their ability, for each model we use the settings recommended by the model distributor. Hyperparameters for each model are reported in Table 4. We use random seeds 0, 123, and 324229 in our experiments.

| Model | Resolution | Steps | Guidance Scale |
|---|---|---|---|
| SD1.5 | 512 | 50 | 7.5 |
| SDXL-Turbo | 512 | 4 | 0 |
| SDXL | 1024 | 50 | 7.5 |
| SD3 | 1024 | 50 | 7.5 |
| FLUX-Time | 1024 | 4 | 0 |

Table 4: Hyperparameters for text-to-image models.

## A.3    VQA WITHOUT V

We use the following prompt for the LLM judge in the VQA without V experiment in Section 4.2:

*You will be given an image description and a question. Your role is to answer the question based on the image description. Image description: [caption]. Question: [question]* where we replace "[caption]" with $F(v)$ and "[question]" with $q$.

To generate the image description, we use the prompt *"Write a caption for this image."* for all image-to-text models.

| Model | # Params | Vision Encoder | Projector | LLM |
|---|---|---|---|---|
| BLIP2-2.7B | 3.8B | EVA-CLIP ViT-g (1.1B) | QFormer | OPT (2.7B) |
| BLIP2-6.7B | 7.8B | EVA-CLIP ViT-g (1.1B) | QFormer | OPT (6.7B) |
| LLaVA1.5-7B | 7.1B | CLIP ViT-L (304M) | MLP | Vicuna-1.5 (7B) |
| LLaVA1.5-13B | 13.4B | CLIP ViT-L (304M) | MLP | Vicuna-1.5 (13B) |
| LLaVA1.6-7B | 7.6B | CLIP ViT-L (304M) | MLP | Mistral (7B) |
| LLaVA1.6-34B | 34.8B | CLIP ViT-L (304M) | MLP | Nous-Hermes-2-Yi (34B) |
| InternVL2-2B | 2.5B | InternViT (304M) | MLP | InternML (2.2B) |
| InternVL2-8B | 8.1B | InternViT (304M) | MLP | InternML (7.7B) |
| InternVL2-26B | 25.5B | InternViT (5.5B) | MLP | InternML (19.9B) |
| InternVL2-40B | 40B | InternViT (5.5B) | MLP | InternML (34.4B) |
| LLaVA-OV-0.5B | 0.9B | SigLIP ViT-L/14 (307M) | MLP | Qwen-2 (0.5B) |
| LLaVA-OV-7B | 8B | SigLIP ViT-L/14 (307M) | MLP | Qwen-2 (7B) |

Table 5: **Summary of image-to-text models on model architecture and scale.**

| Model | # Params | Image Generator | Context Dim. | Text Encoder | Dataset Re-captioning |
|---|---|---|---|---|---|
| SD1.5 | 983M | UNet (860M) | 768 | CLIP ViT-L | ✗ |
| SDXL | 3.5B | UNet (2.6B) | 2048 | CLIP ViT-L & OpenCLIP ViT-G (817M) | ✗ |
| SDXL-Turbo | 3.5B | UNet (2.6B) | 2048 | CLIP ViT-L & OpenCLIP ViT-G (817M) | ✗ |
| FLUX-Time | 12B | MMDiT | 2816 | CLIP ViT-L & T5 XXL | - |
| SD3 | 2B | MMDiT | 4096 | CLIP ViT-L & OpenCLIP ViT-G & T5 XXL | 50% real 50% CogVLM captions |

Table 6: **Summary of text-to-image models on model architecture and scale.**

## B    CYCLE CONSISTENCY FOR ALL MODEL COMBINATIONS

We report cycle reconstruction scores across all different model combinations (13 image-to-text models × 5 text to image models) for both text and image cycles. Figures 14 and 15 display heatmaps of scores for image and text cycle consistency respectively.

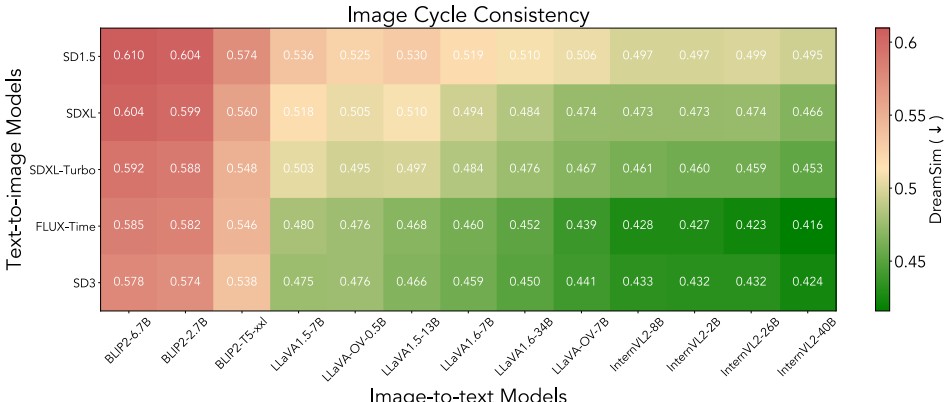

Figure 14: **Image cycle consistency on DCI dataset**. We report the average score across 3 random seeds.

## C    ABLATIONS ON MEASURING CYCLE CONSISTENCY

### C.1    METRICS

For image cycle consistency, we measure DreamSim (Fu et al., 2023b), LPIPS (Zhang et al., 2018), CLIP (Radford et al., 2021), and MSE between input and reconstructed images. Correspondingly, we measure the distance between input text and reconstructed text with various text similarity metrics: BertScore (Zhang et al., 2019), BartScore (Yuan et al., 2021), SBERT (Reimers & Gurevych, 2019), and CLIP (Radford et al., 2021). For CLIP, we measure the cosine similarity between features from the CLIP text encoder. Image-text alignment is measuring using CLIP and ImageReward Xu et al. (2024).

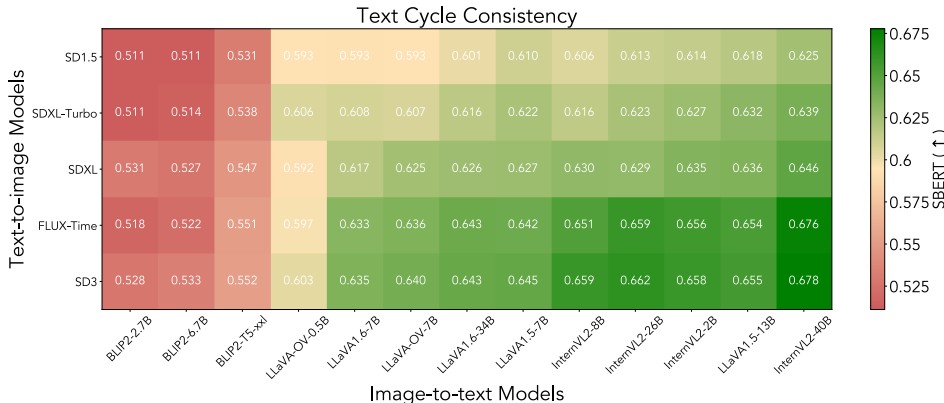

Figure 15: **Text cycle consistency on DCI dataset**. We report the average score across 3 random seeds.

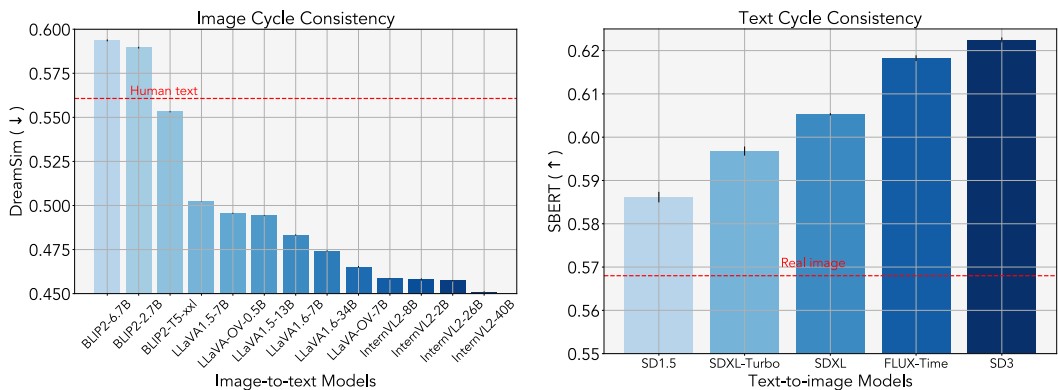

Figure 16: **Human text and real image as baselines.** We compare human text as a baseline to synthetic captions generated by image-to-text models, and real image as a baseline to synthetic images generated by text-to-image models. We detail why real image performs worse in Figure 17.

Similarly to Figure 7, we investigate correlation between cycle consistency and captioning/text-to-image generation quality by ablating different similarity metrics. Tables 7, 8 report the Pearson correlation coefficient between cycle consistency and captioning or image generation performance respectively for different reconstruction metrics. Perceptual similarity metrics, i.e., DreamSim and LPIPS, align best with captioning performance, followed by CLIP and MSE. We additionally compare image-text similarity metrics and find that image reconstruction scores are more predictive of image captioning performance. For MSCOCO images and captions, we also report compare cycle consistency computed with various metrics and POPE in Table 9. Again, DreamSim has the strongest correlation.

## C.2 MODEL ABLATIONS FOR MEASURING CYCLE CONSISTENCY

In this paper, cycle consistency calculations for image-to-text models are averaged across all 5 text-to-image models. Another option would be to choose a text-to-image model to fix, and then use this one fixed model to calculate cycle consistency for all image-to-text models. This section investigates the how this choice of fixed model (or averaging across all fixed models) affects cycle consistency correlations in Figure 7.

We report the Pearson correlation coefficient per model. As shown in Figure xxx, the correlation is consistently strong for most models ($R^2 > 0.65$), except for `BLIP2-2.7B` and `LLaVA-OV-0.5B` with lower coefficients of $0.349$ and $0.241$, respectively. We attribute the low correlation to their use of small-scale, less-performant language models (`OPT-2.7B`, `Qwen2-0.5B`) as pre-trained backbones which may cause poorer text reconstruction.

## D DOWNSTREAM PERFORMANCE

Input Text | Real Image | Reconstructed Text

A group of diverse tents are set up on the ground, surrounded by people, cars, and buildings. Orange cones and logs are scattered on the sidewalk and ground. The buildings in the background are a mix of gray, brown, and trees, with a palm tree on the left. The tents include a red, white, and neon green tent tops.

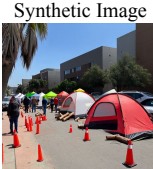

[SBERT 0.683] The image depicts a bustling outdoor event taking place in a park-like setting. The scene is lively and vibrant, with numerous colorful tents and canopies set up in the foreground, indicating a festival or market. The tents are of various colors, including red, green, blue, and white, and are arranged in a somewhat organized manner, suggesting different stalls or booths for

Input Text | Synthetic Image | Reconstructed Text

A group of diverse tents are set up on the ground, surrounded by people, cars, and buildings. Orange cones and logs are scattered on the sidewalk and ground. The buildings in the background are a mix of gray, brown, and trees, with a palm tree on the left. The tents include a red, white, and neon green tent tops.

[SBERT **0.833**] The image depicts an outdoor scene where several tents are set up on what appears to be a paved area, possibly a parking lot or a similar open space. The tents are of various colors, including red, white, and green, and are arranged in a somewhat orderly fashion. The tents are pitched on the ground, and some are supported by wooden logs placed

Figure 17: **Why do synthetic images achieve better text cycle consistency compared to real images?** We visualize text cycle consistency from a real image vs. synthetic image. Compared to real images containing more complex detail, synthetic images only generate details *described in the input text* which occupy larger areas of the generated image. Therefore, such details are easier to reconstruct for the image-to-text model, resulting in better text reconstruction.

| Dataset | Image Cycle Consistency | | | | Image-Text Similarity | |
|---|---|---|---|---|---|---|
| | **DreamSim** | **LPIPS** | **CLIP** | **MSE** | **CLIP** | **Image Reward** |
| MME | 0.705 | 0.475 | **0.806** | 0.247 | 0.748 | 0.427 |
| SEEDBench | **0.728** | 0.794 | 0.621 | 0.431 | 0.587 | 0.067 |
| MMStar | **0.640** | 0.599 | 0.544 | 0.477 | 0.617 | 0.228 |
| Average | **0.833** | 0.820 | 0.788 | 0.692 | 0.831 | 0.265 |

Table 7: **Pearson correlation coefficient between similarity metrics and captioning performance.** Perceptual similarity metrics, such as DreamSim and LPIPS, are the best predictors of captioning performance, followed by CLIP and MSE. DreamSim reconstruction generally shows a higher correlation than both image-text similarity metrics.

| Dataset | Text Cycle Consistency | | | | Text-Image Similarity | |
|---|---|---|---|---|---|---|
| | **BARTScore** | **BERTScore** | **CLIP** | **SBERT** | **CLIP** | **Image Reward** |
| T2I-CompBench | 0.940 | 0.795 | 0.966 | **0.992** | 0.720 | 0.961 |
| DrawBench | **0.945** | 0.894 | 0.832 | 0.861 | 0.465 | 0.814 |
| Average | 0.954 | 0.825 | 0.952 | **0.979** | 0.675 | 0.944 |

Table 8: **Pearson correlation coefficient between similarity metrics and text-to-image performance.** Overall, text cycle consistency strongly correlates with text-to-image quality across all metrics. SBERT shows the highest correlation, followed by BARTScore and CLIP, all of which outperform image-text similarity metrics.

Section 4.2, discusses how captioning performance measured by VQA without V is strongly associated with both image and text cycle consistency. Now, we examine if the same correlation holds for model VQA and downstream performance. For image-to-text models, we examine the relationship between cycle consistency and VQA performance in the table below for both images and text. Note that this is an evaluation of model performance, while VQA without V evaluates the quality of the text generated by the model.

For VQA performance, we use reported scores on benchmarks MMBench Liu et al. (2023c) and MME Fu et al. (2023a). MME is split into perception and cognition categories. Cycle consistency is computed on the sDCI

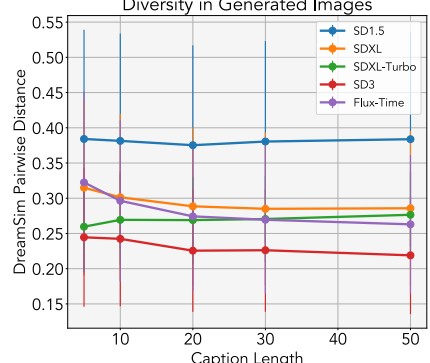

Figure 19: Main Pairwise Distance Between Different Random seeds generated from different caption lengths.

dataset and averaged across five different text-to-image models with 3 different random seeds. Fig-

| Metric | Accuracy | Precision | Recall | F1 Score |
|---|---|---|---|---|
| **Image Cycle Consistency** | | | | |
| DreamSim | **79.10** | **99.32** | **58.66** | **73.76** |
| LPIPS | 77.47 | 99.29 | 55.33 | 71.06 |
| CLIP | 77.99 | 99.18 | 56.44 | 71.94 |
| MSE | 74.64 | 99.07 | 49.74 | 66.23 |
| **Image-text Similarity** | | | | |
| CLIP | 78.04 | 99.30 | 56.49 | 72.01 |
| Image Reward | 75.92 | 99.13 | 52.30 | 68.48 |

Table 9: **Top-1 hallucination and descriptiveness on MSCOCO.** For a given image and a set of corresponding captions, we select the top-1 caption based on each metric. We observe that DreamSim cycle consistency favors captions with less hallucination and better descriptiveness compared to other metrics. Higher precision indicates reduced hallucination, while higher recall reflects increased descriptiveness.

| Fixed I2T Model | ICC $R^2$ | TCC $R^2$ |
|---|---|---|
| BLIP2-2.7B | 0.836 | 0.349 |
| BLIP2-6.7B | 0.834 | 0.657 |
| BLIP2-FlanT5-XXL | 0.879 | 0.871 |
| LLaVA1.5-7B | 0.915 | 0.966 |
| LLaVA1.5-13B | 0.916 | 0.964 |
| LLaVAOV-0.5B | 0.932 | 0.201 |
| LLaVAOV-7B | 0.954 | 0.910 |
| LLaVA1.6-7B | 0.942 | 0.963 |
| LLaVA1.6-34B | 0.953 | 0.952 |
| InternVL2-2B | 0.904 | 0.935 |
| InternVL2-8B | 0.903 | 0.904 |
| InternVL2-26B | 0.880 | 0.879 |
| InternVL2-40B | 0.902 | 0.913 |
| All Models | 0.902 | 0.950 |

| Fixed T2I Model | ICC $R^2$ | TCC $R^2$ |
|---|---|---|
| SD1.5 | 0.741 | 0.875 |
| SDXL | 0.759 | 0.845 |
| SDXL-Turbo | 0.731 | 0.879 |
| SD3 | 0.790 | 0.870 |
| FLUX-Time | 0.794 | 0.861 |
| All Models | 0.766 | 0.864 |

Table 10: Pearson correlation coefficients between text-to-image performance and image cycle consistency (**ICC**) and text cycle consistency (**TCC**) for different fixed image-to-text models (**Left**) and different fixed text-to-image models (**Right**). Correlations are generally strong except when fixing BLIP2-2.7B or LLaVAOV-0.5B to calculate text cycle consistency. Higher $R^2$ value indicates stronger correlation.

ures 18 shows plots of image and text cycle consistency vs VQA performance on each benchmark, with points representing different image-to-text models.

We find that image cycle consistency best correlates with VQA performance on MMBench and MME (cognition), with weaker association for other benchmarks. This may be somewhat surprising because MME (perception) in the VQA without V setting has a strong correlation with cycle consistency. Text cycle consistency did not have as strong as a correlation across all VQA benchmarks. It is important to note that VQA scores examine if the model can answer diverse questions about an image, while VQA without V examines if the text caption can answer diverse questions about the image. While these two evaluations are somewhat related, the difference between these tasks could account for the difference in correlation.

# E DIVERGENCE IN GENERATED IMAGES

We use the DCI summarized captions dataset Huh et al. (2024) Urbanek et al. (2024) detailed in Section 4.4 to compare the diversity of synthetic images generated from text based on the caption density use to create them. For each image in the summarized DCI dataset, there are summary captions of different length. For each image, we generate generate 10 different images using random seed sampling for every summary length. We then calculate the mean-pairwise distance between all of the 10 generated images from the same summary caption. For each text-to-image model, we plot

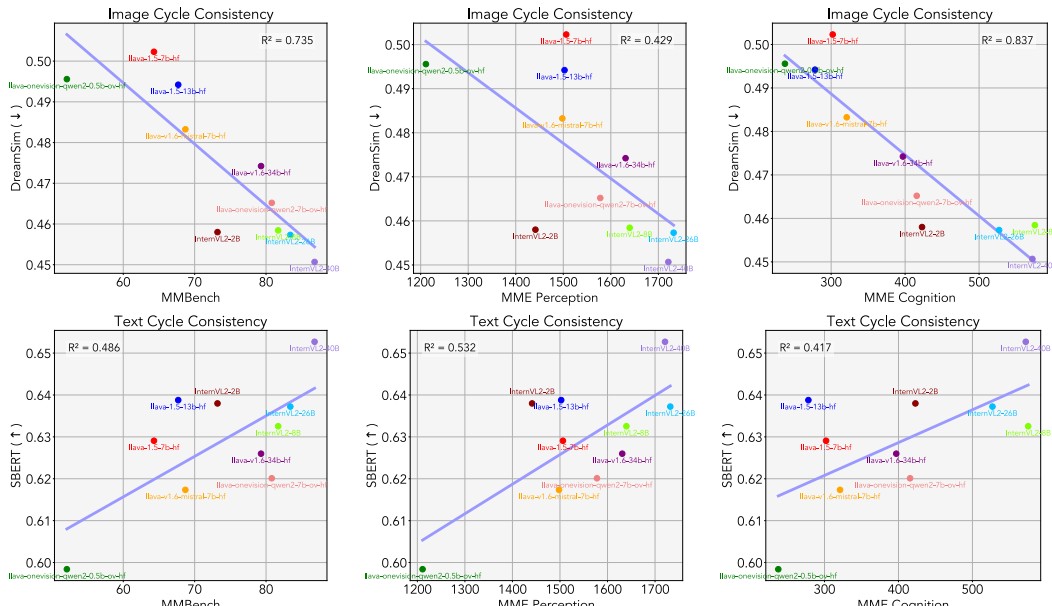

Figure 18: **Cycle consistency vs. Model Downstream Performance (VQA): Top Row (Left to Right)**: Image cycle consistency association with VQA scores on MMBench, MME (Perception), and MME (cognition). **Bottom Row (Left to Right)**: Text cycle consistency association with VQA scores on MMBench, MME (Perception), and MME (cognition).

the mean-pairwise distance vs. the caption length seen in Figure 19. Considering all models, there is mixed consensus on how caption length affects generated image diversity.

# F    ADDITIONAL RESULTS

We show additional "plug and play" examples of image and text cycle consistency in Figures 20, 21 respectively. Figure 22 provides more examples of increased text details hurting image reconstruction scores.

# G    FAILURE CASES

We provide examples of failure cases of cycle consistency in Figures 23 and 24. Failures include: synthetic images with artifacts or implausible generations but little effect on captions, descriptions of non-existent objects, endpoint model failures (i.e. the intermediate image or text representation is reasonable but the endpoint model creates inaccuracies which affect reconstruction). Many of these mistakes can be attributed to model error and usually affect text cycle consistency much more than image, mainly because images generated from incorrect captions often have lower cycle consistency, whereas image-to-text models do not always notice inaccuracies in synthetic images.

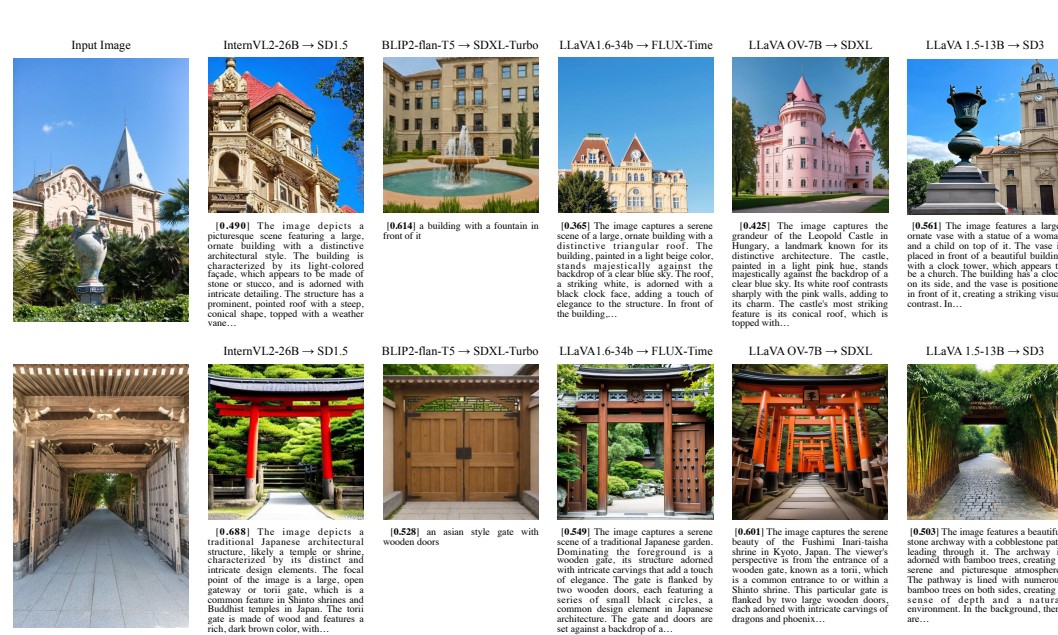

Figure 20: **Examples of image cycle consistency** using different image-to-text, text-to-image combinations. The model combination used to generate the caption → image is shown at the top of the image. DreamSim(↓) distance between reconstructed images and the original are reported in brackets in front of the text captions.

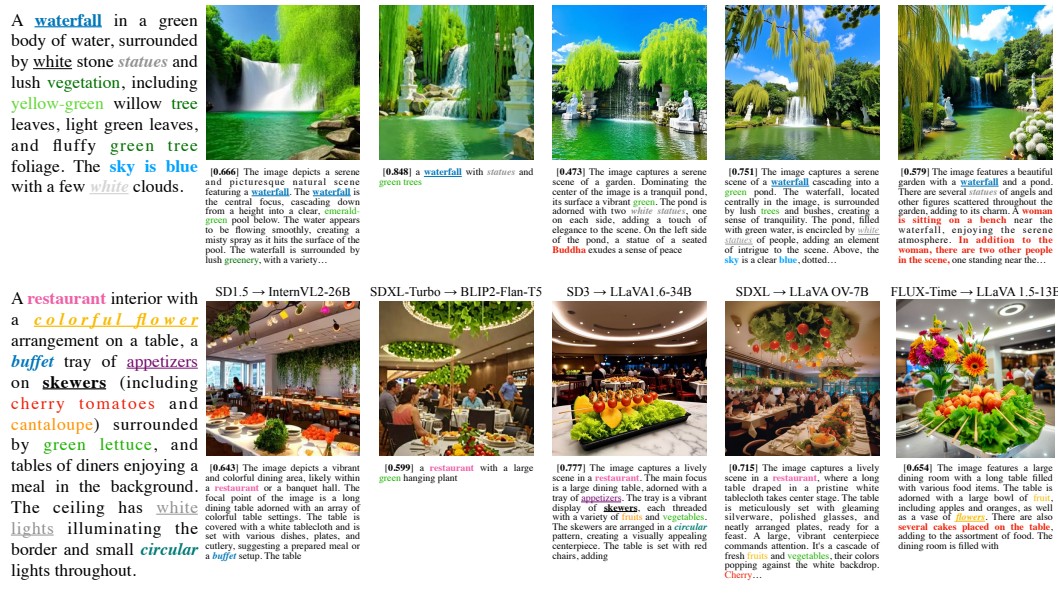

Figure 21: **Examples of text cycle consistency** from various model combinations. The model combination used to generate the image → text caption is shown at the top of each example's image. To make comparing the text easier, we highlight relevant phrases in the input and reconstructed descriptions.

| Input Image | 5 Word Summary | 30 Word Summary |
|---|---|---|

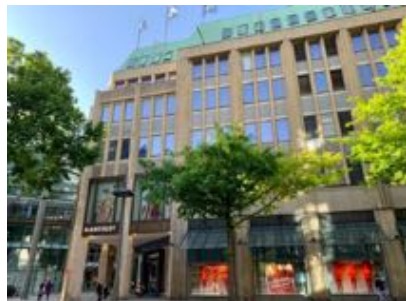 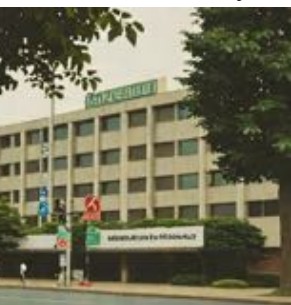 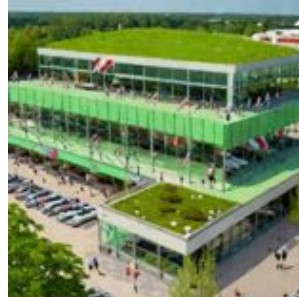

[**0.489**] Large building with trees and signs.

[**0.673**] A large shopping center building with a light green roof, featuring multiple window displays, flags, and a glass and metal awning, surrounded by trees and people walking around it.

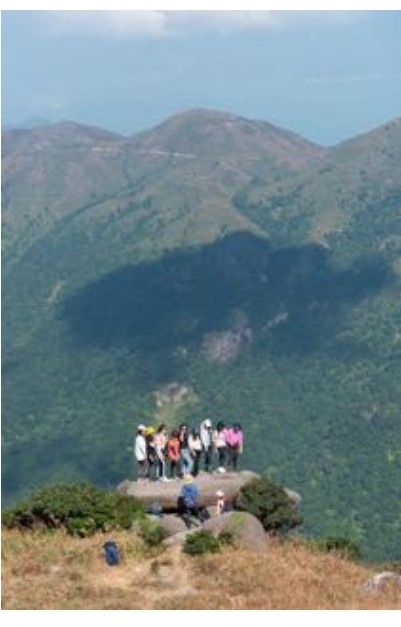 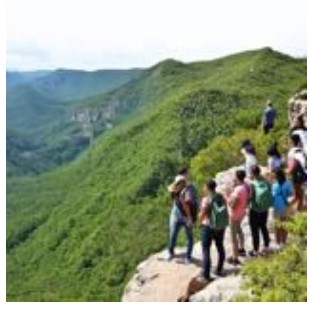 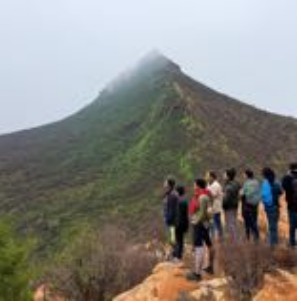

[**0.349**] People on cliff overlooking green mountain.

[**0.480**] A group of people stand on a cliff overlooking a huge green and brown mountain, surrounded by shrubs and dry grass, with a misty sky above.

Figure 22: Examples where increasing the caption detail does decreases image cycle consistency.

| Input Image | SDXL-Turbo [**0.496**] | FLUX-Time [**0.392**] |
|---|---|---|

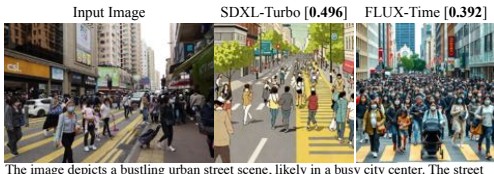

The image depicts a bustling urban street scene, likely in a busy city center. The street is filled with numerous pedestrians, many of whom are wearing face masks, suggesting a concern for health and safety, possibly due to a pandemic. The people are crossing the street at a pedestrian crossing marked with yellow lines. The crowd includes a mix of adults and children, with some pushing strollers

**Input text**: A sleek, silver skyscraper stands prominently in a cityscape, surrounded by trees and water. Its curved design and pointed top make it stand out, and its many windows and horizontal lines indicate multiple floors. A clear blue sky serves as the backdrop, with a slight gradient towards the horizon. Other buildings can be seen in the distance. (Daytime)

**InternVL2-40B [0.797]**: The image depicts a tall, modern skyscraper with a sleek, cylindrical design, standing prominently against a clear blue sky. The building has a reflective metallic exterior, giving it a shiny, mirror-like appearance. The structure tapers slightly towards the top, culminating in a sharp, pointed spire. The skyscraper is surrounded by a body of water in the…

**LLaVA 1.6-34B [0.551]**: The image captures the iconic Petronas Twin Towers, a renowned landmark in Kuala Lumpur, Malaysia. The towers, constructed from steel and glass, stand tall against the backdrop of a clear blue sky. The perspective of the image is from a low angle, looking up at the towers, emphasizing their height and grandeur. The towers are connected by a skybridge…

Generated Image

Figure 23: **Endpoint model failure cases. Left**: Example of lower image cycle consistency measured with DreamSim(↓). Given the same synthetic text generated for an input image, the reconstructed image quality can be very different between models due to errors or discrepancies. **Right**: Example of text reconstructions measured with SBERT(↑) due varying image-to-text model quality. While the synthetic image generated from the text seems reasonable, InternVL2-8B interprets the image correctly while the text from LLaVA 1.6-34B contains several errors leading to poor text cycle consistency.

**Input text**: A **waterfall** in a **green body of water**, surrounded by **white stone statues** and **lush** **vegetation**, including **yellow-green willow tree** leaves, **light green** leaves, and fluffy **green tree** foliage. The **sky is blue** with a few **white** clouds.

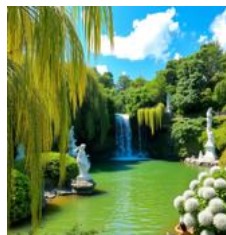

**LLaVA 1.5-13B** [**0.579**]: The image features a beautiful **garden** with a **waterfall** and a pond. There are several **statues** of angels and other figures scattered throughout the **garden**, adding to its charm. A **woman is sitting on a bench** near the **waterfall**, enjoying the serene atmosphere. **In addition to the woman, there are two other people in the scene,** one standing near the…

**InternVL2-2B** [**0.756**]: The image depicts a serene and picturesque **garden** scene featuring a **waterfall** and several **statues**. The garden is **lush** and **verdant**, with a variety of **greenery and plants** surrounding the **waterfall**. The **waterfall** is a central feature, cascading down a rocky ledge into a **pool** below. The **water** appears to be a vibrant **green**, suggesting it might be a natural

Generated Image

**Input text**: A **red-roofed**, **yellow**-white **building** stands in the background, with a **balcony** and **rectangular windows**. In the foreground, a **tree** and **grassy area** with a small **fence** are visible, along with a **black car** parked in a **space**, its doors open and a man nearby. The **sky is light blue** with **white clouds**.

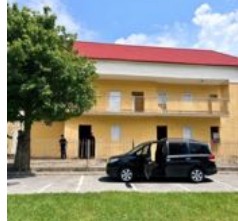

**BLIP2-FlanT5-XXL** [**0.628**]: a **yellow** **building** with a **red roof**

**LLaVA 1.6-34B** [**0.521**]: The image features a **black car** parked in front of a **hotel**, with **a man standing outside** the vehicle. The car is parked in a parking **space**, and the man appears to be either getting out of the **car** or walking towards it. The hotel is a two-story **building** with a **red roof**, and there are several **windows** visible on both floors.

Generated Image

Figure 24: **Failure cases for text cycle consistency.** Text reconstruction is often affected by image captioning errors (top), and hallucinations and failures of the metric to address this (bottom).

