# OpenReview forum: "On the Cycle Consistency of Image-Text Mappings"
_ICLR.cc/2025/Conference — Submitted to ICLR 2025_

### Official Review · Reviewer_BHgf · 2024-10-29

**Soundness:** 3
**Presentation:** 2
**Contribution:** 2
**Rating:** 5
**Confidence:** 4

**Summary:**

The paper (Cycle Consistency of Image-Text Mappings) investigates the degree to which the image-text mappings have a cyclical consistency. Although existing models do not train for this consistency explicitly, for a subset of models this cyclic consistency is enforced. In terms of application, the authors find that the measure of cycle-consistency correlates relatively well with downstream accuracy — which can help perform quick tests on the capabilities of the model without requiring a curated benchmark.  Overall, I believe that the paper is insightful, but lacks a strong application using those insights except for an approximate performance check for downstream tasks.

**Strengths:**

Below I state the strengths and weaknesses of the model:

Strengths:

- The paper is a thorough empirical study on the cycle-consistency of image-text representations across most of the popular T2I and I2I models. The results though not super surprising (as they are often trained / fine-tuned on the similar training sets) are well documented and can be crucial for the community.
- The observation regarding the correlation between image-text cyclical consistency and downstream task accuracy can be useful to quickly check the effectiveness of the model.  One question regarding this: In Fig. 4, SDXL-Turbo is used as a decoder for the image reconstruction case and LLaVa-1.5-13B for text generation. How does this design choice affect the correlation between cycle consistency and downstream performance? The authors should ideally provide some ablation on this design choice.

**Weaknesses:**

Weaknesses:
- The application of using cycle-consistency as an approximate measure for downstream task accuracy is an interesting use-case; However, I believe they are proxy for only two tasks (Image captioning and T2I generation performance). To be useful in practice, I suggest the authors to add in more tasks concerning these models (e.g., VQA amongst others) and check if cycle-consistency can still be an approximate measure of task accuracy.
- I find the Sec.(5) to be intriguing, but the authors should highlight how some of the takeaways be used towards training / fine-tuning models with better downstream capabilities.
- The authors select CIDEr score for captioning performance measurement; Have the authors considered using a strong MLLM for measuring captioning performance and using it to measure the correlation with?
- (This is not a weakness - but a discussion point) — Based on the insights, what do the authors think about building unified image-to-text and text-to-image models while enforcing cyclical consistency? Will it lead to better downstream performance than training these models independently.

**Questions:**

Overall, this paper is a nice empirical study on cyclical consistency of image-text mappings, but I would urge the authors to respond to the Weaknesses during the rebuttal. I am open to improving the score based on the rebuttal discussion. Looking forward to the discussion.

See Weaknesses for additional questions.

---

> ### Author Response · Authors · 2024-11-28
> **Individual response to BHgf**
>
> Thank you for the insightful feedback and the helpful suggestions.
>
> ### Q1. How does model choice affect the correlation between cycle consistency and downstream performance?
> As requested, we report the Pearson correlation coefficient **per model**. The **correlation is consistently strong** for most models ($R^2 > 0.65$), except for BLIP2-2.7B and LLaVA-OV-0.5B with lower coefficients of 0.349 and 0.241, respectively. We attribute the low correlation to their use of small-scale, less-performant language models (OPT-2.7B, Qwen2-0.5B) as pre-trained backbones, which may cause poorer text reconstruction.
>
> | Fixed I2T Model | Text Cycle Consistency vs T2I Model Performance ($R^2$) |
> |-----------|-----------|
> | BLIP2-2.7B | 0.349 |
> | BLIP2-6.7B | 0.657 |
> | BLIP2-T5-xxl | 0.871 |
> | LLaVA1.5-7B | 0.966 |
> | LLaVA1.5-13B | 0.964 |
> | LLaVA-OV-0.5B | 0.201 |
> | LLaVA-OV-7B | 0.910 |
> | LLaVA1.6-7B | 0.963 |
> | LLaVA1.6-34B | 0.952 |
> | InternVL2-2B | 0.935 |
> | InternVL2-8B | 0.904 |
> | InternVL2-26B | 0.879 |
> | InternVL2-40B | 0.913 |
> | Average | 0.950 |
>
> |Fixed I2T Model| Image Cycle Consistency vs T2I Model Performance  ($R^2$) |
> |-----------|-----------|
> | BLIP2-2.7B | 0.836 |
> | BLIP2-6.7B | 0.834 |
> | BLIP2-T5-xxl | 0.879 |
> | LLaVA1.5-7B | 0.915 |
> | LLaVA1.5-13B | 0.916 |
> | LLaVA-OV-0.5B | 0.932 |
> | LLaVA-OV-7B | 0.954 |
> | LLaVA1.6-7B | 0.942 |
> | LLaVA1.6-34B | 0.953 |
> | InternVL2-2B | 0.904 |
> | InternVL2-8B | 0.903 |
> | InternVL2-26B | 0.880 |
> | InternVL2-40B | 0.902 |
> | All Models | 0.924 |
>
> |Fixed T2I Model| Text Cycle Consistency vs I2T Model Performance ($R^2$) |
> |-----------|-----------|
> | SD1.5 | 0.875 |
> | SDXL-Turbo | 0.845 |
> | SDXL | 0.879 |
> | SD3 | 0.870 |
> | FLUX Time | 0.861 |
> | All Models | 0.864 |
>
> |Fixed T2I Model| Image Cycle Consistency vs I2T Model Performance ($R^2$) |
> |-----------|-----------|
> | SD1.5 | 0.741 |
> | SDXL-Turbo | 0.759 |
> | SDXL | 0.731 |
> | SD3 | 0.790 |
> | FLUX Time | 0.794 |
> | All Models | 0.766 |
>
> Furthermore, we have updated Figures 6, 7, 10 results to report cycle consistency **averaged across all models**, instead of just fixing one model in the pipeline. We also extend the analysis to **include all four combinations**, additionally comparing text quality (descriptiveness, hallucination) and image quality (prompt-following) with both image and text cycle consistency.  We observe that both cycles exhibit a **strong correlation** across modalities, with text cycle consistency being more prominent.
>
>
> ### Q2. Add more tasks (e.g., VQA) and check if cycle consistency can be a proxy of task accuracy.
> As requested, we report correlation between cycle consistency and VQA performance. Note that this is an evaluation of **model** performance, whereas VQA without V measures the descriptiveness of **text descriptions**. We evaluate VQA performance on MMBench and MME, with MME divided into perception and cognition categories. Cycle consistency is computed on the sDCI dataset and averaged across five different text-to-image models with 3 different random seeds. The table below reports the Pearson correlation coefficient ($R^2$) between VQA scores and cycle consistency. Image cycle consistency strongly correlates with MMBench and MME (cognition), but weaker with other benchmarks. This is somewhat surprising as MME (perception) shows a strong correlation with cycle consistency in our VQA without V experiment. This discrepancy likely stems from the differences in these tasks: VQA evaluates the **model**’s ability to answer diverse questions about an image, while VQA without V evaluates whether the **text descriptions** include detailed answers to those questions. We have also added the correlation plots to the Appendix as Figure 18.
> | Dataset | Image Cycle Consistency | Text Cycle Consistency |
> |-----------|-----------|-----------|
> |  MMBench | 0.735 | 0.486 |
> | MME (average) | 0.363 | 0.553 |
> | MME (perception) | 0.429 | 0.532 |
> | MME (cognition) | 0.837 | 0.417 |

---

> ### Author Response · Authors · 2024-11-28
> **Individual response to BHgf (2)**
>
> ### Q3. How Section 5 takeaways can be used for model training.
> It is an interesting question how to incorporate the results of Section 5 (sensitivity to prompt style) for better model training. Because image-text mappings are one-to-many (or even many-to-many), some variance is expected and even desirable for sampling over different outputs. The degree to which two text prompts are equivalent could be even a model design choice if introduced as a type of augmentation during training, so that different sentences with the same sentiment map to the same images (this is similar to image augmentation for vision models).
>
> Note that we have extended our analysis to study how **random seed selection, prompt and caption style, and temperature sampling** contribute to variance in cycle consistency (updated Section 5). Table 2 shows that image-to-text models exhibit higher variance due to temperature sampling but remain relatively robust to changes in prompt style. In contrast, text-to-image models are significantly more sensitive to caption style than to random seed sampling.
>
>
> ### Q4. Using a strong MLLM for measuring captioning performance.
> We clarify that we use a strong LLM for measuring captioning performance instead of CIDEr score. We use a VQA Benchmark dataset (such as MME, MMStar) and ask each image-to-text model to produce descriptions for the images in the benchmark dataset. Then, we ask an LLM to answer the VQA question based on the output text descriptions (instead of the images, which would be traditional VQA). This measure for captioning performance has several advantages over CIDEr scores. Firstly, CIDEr scores compare caption outputs against human annotated captions, which are often short and lack detail. Secondly, we are able to determine what kind of information are included in different captions, and what aspects image-to-text models are better at describing than others (based on the VQA sub-categories). Figure 7 demonstrates that cycle consistency strongly correlates with descriptiveness in the generated captions.
>
> ### Q5. What do the authors think about building unified image-to-text and text-to-image models while enforcing cycle consistency?
> Based on our findings, along with evidence of existing high-performing models which **already** incorporate cycle consistency in training [1-4], we believe that enforcing cycle consistency can be a helpful training objective. For text-to-image models, DALLE-3 [1] and SD3 [2] are trained on descriptive captions generated by an image captioning model. This process can be formalized as $\text{argmin}_G \ L(I, G(F(I)))$ where $I$ is the input image, $F$ is the image-to-text model, and $G$ is the text-to-image model. Training a text-to-image model on synthetic captions is equivalent to enforcing image cycle consistency relative to the fixed image captioner. For vision-language models (VLMs), Synth2 [3] trains a VLM using data from a pre-trained text-to-image model, while ITIT [4] jointly trains image-to-text and text-to-image models to be cycle-consistent. By injecting cycle consistency during training, both Synth2 and ITIT achieve high performance with significantly fewer data examples than state-of-the-art models. We have included this discussion in the introduction (Section 1) to further motivate our analysis of cycle consistency in current models.
>
> [1] Improving image generation with better captions. Betker et al., 2023.
>
> [2] Scaling rectified flow transformers for high-resolution image synthesis. Esser et al., ICML 2024.
>
> [3] Synth2: Boosting visual-language models with synthetic captions and image embeddings. Sharifzadeh et al., 2024.
>
> [4] Leveraging unpaired data for vision-language generative models via cycle consistency. Li et al., ICLR 2024.

---

### Official Review · Reviewer_oP89 · 2024-10-30

**Soundness:** 2
**Presentation:** 3
**Contribution:** 1
**Rating:** 3
**Confidence:** 4

**Summary:**

In this work, the authors explored about the multi-modal cycle consistency of current text-to-image(T2I) and image-to-text(I2T) generation models.

They paired various off-the-shelf T2I and I2T models to build a cycle model and measured the input-output difference.  They found that current state-of-the-art models possess a certain level of perceptual cycle consistency, even when they're not explicitly trained with cycle consistency objectives.
Then, they argued that as the performance of the individual T2I/I2T module increases, the cycle consistency improves.

To further analyze and find possible factors that can affect to achievement of cycle consistency, the authors suggested the concept of 'divergence' in T2I mappings. And they claimed that more detailed and informed text prompts showed more divergent output space, yet improved cycle consistency.
Finally, the authors demonstrated that a slight perturbation of text input sometimes results in higher variation in the T2I model output, which could be a challenge to achieve better cycle consistency.

**Strengths:**

-Considering the current generative models become more challenging to inject cycle consistency because of their iterative sampling process, their behavior on cycle consistency is an interesting question.

-The script is well-written and clearly presents its claim.

**Weaknesses:**

-Some experiments are not well-designed, which makes the corresponding findings seem to lack contributions or doubtful.

1. Section 3 stated to demonstrate "more cycle-consistent models have better T2I/I2T downstream performance",  but its content only shows that "better T2I/I2T models are more cycle-consistent", which are not the same.
It seems too natural that combining better T2I&I2T models improves cycle-consistency of the pair, since they provide high-quality data that contains major information of the input. On the other hand, it's still questionable that satisfying cycle consistency guarantees better T2I&I2T performance.
(e.g. A perfect Image->Text->Image reconstruction can be achieved if the I2T model writes down all pixel values in one long string. A perfect Text->Image->Text reconstruction can be achieved if the T2I model produces the image that contains the entire input text visually.)

2. In Figure 6, synthesized input captions with fewer tags don't seem to actually contain less information. In the first row, the input caption for 1 Tag is very long and specific, more detailed than 2~5 Tag captions. In the second row, the 1 Tag caption already contains the info of the second tag "reflects". This could be the reason that the divergence decreased with fewer tags, since better cycle consistency (more tags) coming with more divergence seems counter-intuitive.

**Questions:**

-In Table 1\~2, the presented values alone are not enough to tell if each I2T+T2I model pair has a good cycle consistency since there's no baseline performance or threshold was suggested. Although the authors showed several cases in Figure 2\~3, could the authors provide any kind of baseline scenario for comparison?

-Since the sampling process image-to-text models can be also stochastic, could you also provide the analysis on the divergence of I2T models?

-What does the analysis of the divergence and sensitivity of I2T models suggest for creating more cycle-consistent models? It would be better if there's a clearer statement about how the results on divergence and sensitivity imply about cycle consistency.

---

> ### Author Response · Authors · 2024-11-28
> **Individual response to oP89**
>
> Thank you for the insightful feedback and the helpful suggestions.
>
> ### Q1. It is questionable that satisfying cycle consistency guarantees better performance.
> Thank you for pointing this out and we will clarify this in the paper. We agree that cycle consistency is likely a consequence of better model capabilities – updated Section 3 analyzes contributing factors for enhanced cycle consistency. We are **not** suggesting that cycle consistency is necessarily a contributing factor to performance, unless models are known to incorporate cycle consistency during their training (e.g., SD3 and DALLE-3).
> > On the other hand, it's still questionable that satisfying cycle consistency guarantees better T2I&I2T performance. (e.g. A perfect Image->Text->Image reconstruction can be achieved if the I2T model writes down all pixel values in one long string. A perfect Text->Image->Text reconstruction can be achieved if the T2I model produces the image that contains the entire input text visually.)
>
> While we agree that there are trivial solutions to the I2T2I and T2I2T cycles, we do not encounter a failure mode to this degree, since it is unlikely that models are trained with these kind of examples (i.e., outputting pixel values in text or generating text descriptions in pixels). Instead, we demonstrate that higher cycle consistency is generally achieved by higher-quality generated text and images, i.e., more descriptive text with less hallucinations and images generated faithfully to the text and containing fine-grained details (see updated Section 4).
> Apart from the rare scenario suggested by the reviewer, we demonstrate in Section 5 that cycle consistency is sensitive to subtle perturbations in language (i.e., prompt style). Additionally, Section 4.2 (Figure 9) shows instances where high image cycle consistency is obtained using short captions lacking details. While such cases are not uncommon for image cycle consistency, we do not observe similar cases in text cycle consistency.
>
> ### Q2. How divergence and sensitivity analysis relate to cycle consistency. Fewer tags don’t necessarily contain less information.
> We agree that previous Section 4 (divergence) lacks proper control of caption length and information - instead it varies the number of elements described. Furthermore, Section 5 (sensitivity) combined changes in length and style during prompt rewriting, making it difficult to isolate their respective impacts on cycle consistency. To address these shortcomings and better relate Sections 4 and 5 to cycle consistency, we make the following changes:
> 1. Updated Section 4.4 studies **cycle consistency as a function of caption length**. Captions from the Densely Captioned Images dataset [2] are summarized into varying lengths (5, 10, 20, 30, and 50 words) using LLaMA3-8B-Instruct. Figure 12 shows that cycle consistency improves as captions become more descriptive, especially for the higher performing models FLUX-Time and SD3.
> 2. Updated Section 5 studies the **variance in cycle consistency** (formerly called divergence). Unlike the previous experiment, we address the effect of caption length separately in Section 4.4, and focus on sources of variance in this section. Specifically, we analyze how random seed selection, prompt style, and temperature sampling contribute to this variance. Table 2 shows that image-to-text models exhibit higher variance due to temperature sampling but remain relatively robust to changes in prompt style. In contrast, text-to-image models are significantly more sensitive to prompt style than to random seed sampling.
> 3. Congruent with the tags experiment, we report **diversity as a function of caption density**, measured by DreamSim pairwise distance between generated images using 10 different random seeds for the same caption. However, we observe inconsistent trends across the text-to-image models (see updated Figure 19). This discrepancy likely stems from differences in the experiment design: previously Section 4 used hierarchically created captions with tags, often altering meaning by introducing new elements, while the caption density experiment used summarized versions of the same caption with varying levels of detail. By focusing on caption density rather than tags, we aim to better understand how caption descriptiveness influences cycle consistency.

---

> ### Author Response · Authors · 2024-11-28
> **Individual response to oP89 (2)**
>
> ### Q3. Baselines for cycle consistency.
> We add the following baselines and comparisons in Figure 16:
> 1. For **image cycle consistency**, we compare against reconstructing an image given **human annotated text**. For this baseline, each image in the DCI dataset is paired with a “short caption” provided by a human annotator. We compare $\text{DreamSim}(I, G(T_\text{human}))$ with $\text{DreamSim}(I, G(F(I))$, comparing human text $T_\text{human}$ against generated text F(I), where $G$ is the image-to-text model and $G$ is the text-to-image model. Figure 16 shows that synthetic text surpasses human text beyond a certain point due to superior descriptiveness, highlighting its effectiveness as a substitute for human text in training large models.
> 2. For **text cycle consistency**, we compare against reconstructing text given a **natural image**. Specifically, we compare $\text{SBERT}(T, F(I))$ with $\text{SBERT}(T, F(G(T))$, where $I$ is the real image paired with the input text $T$. Figure 16 shows that synthetic images achieve better text cycle consistency compared to real images. Figure 17 visualizes text cycle consistency from a real image vs. synthetic image. Compared to real images containing more complex details, synthetic images only generate details described in the input text which occupy larger areas of the generated image. Therefore, such details are easier to reconstruct for the image-to-text model, resulting in better text reconstruction.
>
>
> ### Q4. Analysis of the divergence of image-to-text models.
> As mentioned in **Q2**, the updated Section 5 studies the variance in cycle consistency (formerly called divergence). Specifically, analysis on how 1) **prompt style** and 2) **temperature sampling** affects variance in cycle consistency relates to variance from **image-to-text models**.
>
> [1] Cyclegan, a master of steganography, Chu et al., NIPS “Machine Deception” Workshop, 2017.
>
> [2] A Picture is Worth More Than 77 Text Tokens: Evaluating CLIP-Style Models on Dense Captions, Urbanek et al., CVPR 2024.

---

> > ### Comment · Reviewer_oP89 · 2024-11-29
> >
> > I'd like to thank the authors for their responses, and I appreciate that they resolved many of my questions.
> >
> > However, I still have several concerns:
> >
> > 1. If the authors do not seek cycle consistency to accomplish better T2I/I2T mapping, then why should future researchers take cycle consistency into account? Most of the previous works that involved cycle consistency[1][2] used it as a means for better mapping (between text/image or two different image domains, etc.). If this is not what the authors want to show, what would be the benefit of good cycle consistency? Are there scenarios such that cycle consistency is beneficial as itself?
> >
> > 2. I'm afraid that the manuscript has deviated too far from the first draft; almost every figures and table were re-drawn, experiments in Sections 4 and 5 were done with different control factors and different experiment designs, and some of them came up with different conclusions from the original manuscript. Although I agree that these new contents strengthen the author's idea, I'm concerned that this drastic revision might be against the purpose of the original submission deadline.
> > ---
> > [1] Unpaired Image-to-Image Translation using Cycle-Consistent Adversarial Networks, ICCV 2017
> >
> > [2] Leveraging Unpaired Data for Vision-Language Generative Models via Cycle Consistency, ICLR 2024

---

> > > ### Author Response · Authors · 2024-12-03
> > > **Individual response to oP89 (3)**
> > >
> > > ### Q1. Benefits of cycle consistency.
> > > We highlight the benefit of cycle consistency in Section 4: cycle-consistent mappings **strongly correlate** with **improved descriptiveness and reduced hallucination in generated text, and better prompt-following in images**, i.e., several desired properties when building a proficient multimodal model. Because cycle consistency aligns well with performance, it can be used as a **self-supervised** proxy for such performance measures. Furthermore, cycle consistency allows us insights into what kinds of texts and images are easily exchangeable, and what kinds of data are harder to translate.
> > >
> > > ### Q2. Change in manuscript.
> > > We have mainly updated the manuscript to better communicate the results **requested by the reviewers**. However, we emphasize that the core topic, main results, and conclusions of the paper remain unchanged:
> > > * Figures 6 and 7: Trends remain the same, but we average across **all model combinations** and plot against both cycles to address concerns from Reviewers QcCd and BHgf.
> > > * Table 2: We plot against **cycle consistency** rather than diversity to address concerns from Reviewers QcCd and oP89.
> > > * Figure 12: Addresses concerns from Reviewers 31TL and oP89.
> > > * Table 1, Figure 4: Addresses concerns from Reviewers 31TL, go1x, and QcCd.
> > > * We also improved plot design and added qualitative visualizations to **enhance the quality** of the manuscript.
> > >
> > > We found the questions raised by the reviewers to be highly meaningful, accompanied by its results, which led to a **reorganization** of the sections. We are sincerely grateful for these insightful suggestions, which significantly enhanced the depth and quality of our analysis.

---

### Official Review · Reviewer_31TL · 2024-11-02

**Soundness:** 2
**Presentation:** 3
**Contribution:** 2
**Rating:** 3
**Confidence:** 4

**Summary:**

This paper presents several intriguing phenomena regarding the cycle consistency of image-text mappings with text-to-image models and image-to-text ones. It demonstrates (1) that more advanced models achieve better cycle consistency; (2) a strong correlation between cycle consistency and tasks such as image captioning and text-to-image generation; (3) that the number of objects described by the text affects text-to-image mappings; and (4) that text-to-image models are sensitive to prompt variations.

**Strengths:**

- The paper is well presented and easy to read.
- The paper demonstrates extensive experiments incorporating multiple combinations of T2I and I2T models.

**Weaknesses:**

Major issues:
- Regarding Table 1 and 2, the analysis of why image-to-text models have greater impact than text-to-image models is hand-wavy. There should be more discussion on this.
- Causality Direction: While the paper demonstrates correlation between cycle consistency and model performance (captioning/generation), it fails to address the causality direction. The improved cycle consistency is likely a consequence of better model capabilities rather than a contributing factor, which diminishes the practical utility of cycle consistency as a metric.
- The claim that "more descriptive text leads to a a broader distribution of generated images" is not convincing. The experiments does not properly control the caption length. Figure 6 shows a case where the 1-tag caption exceeds the 5-tag caption in length.
- The abstract makes several claims that aren't supported by the paper's content:
  - "analyze in what ways they (cycle consistency) fail": there are no such discussions in the paper.
  - "how it affects achieving cycle consistency": there are no such discussions in the paper.
  - "we show possible challenges of training cycle consistent models due to the sensitivity of text-to-image models": there are no explorations of training cycle-consistent models in the paper.

Minor issues:
- "more descriptive text leads to a a broader distribution of generated images" has double "a".
- On the 4th line in page 4, the sentence "Therefore, examine how a text-to-image mapping can diverge from one fixed text prompt into many different images." is incomplete.
- At the end of page 8, "Table 5" should be "Table 6".

**Questions:**

- Is image-text cycle consistency a meaningful metric for model development? Should improving cycle consistency be a priority for model designers? What are the concrete applications or benefits of enhanced cycle consistency?

---

> ### Author Response · Authors · 2024-11-28
> **Individual response to 31TL**
>
> Thank you for the insightful feedback and the helpful suggestions.
>
> ### Q1. Analysis is hand-wavy for Table 1 and 2.
> As suggested by Reviewers go1x, QcCd, and 31TL, we have included an analysis of factors contributing to cycle consistency in the updated Section 3. The main findings are highlighted as follows:
> 1. **Cycle consistency improves with LLM scale**. An image-to-text model consists of a vision encoder, a projector, and a large language model (LLM). Scaling the vision transformer (ViT) for the vision encoder is reported to enhance performance [1], yet a simple MLP projection remains the dominant approach [2-5]. Since no model provides open-sourced weights with varying vision encoder scales, we focus our analysis on ablating the scale of the LLM. Figure 4 demonstrates that scaling the LLM enhances image and text cycle consistency across all image-to-text model families. Figure 5 visualizes this effect—despite being trained on the same dataset and architecture, only InternVL2-40B successfully captures both the color and the presence of a corner turret.
> 2. **Cycle consistency improves with re-captioned dataset quality**. Table 1 demonstrates that the quality of the re-captioned dataset (e.g., dataset re-captioned by GPT-4V, LLaVA1.6-34B) plays an important role in improving image cycle consistency, often outperforming models trained on larger datasets annotated by less-performant models (e.g., BLIP). On the other hand, text cycle consistency shows little difference between the LLaVA models, as the input text from sDCI often lacks fine-grained detail (evidenced in Figure 16) compared to longer and more descriptive synthetic captions, such as those produced by LLaVA1.6 and LLaVA-OV. We believe higher-quality human annotations and text-to-image models with longer context would enhance the analysis of text cycle consistency. We exclude InternVL2 from this analysis as its pre-training dataset details are not disclosed.
> We detail differences in architecture, scale, and dataset in Table 1, 5, and 6.
>
> ### Q2. Failure to address causality.
> We agree that cycle consistency is likely a consequence of better model capabilities – updated Section 3 analyzes contributing factors for enhanced cycle consistency. We are **not suggesting that cycle consistency is a contributing factor to performance**.
>
> However, this does not diminish the practical utility of cycle consistency as a tool to study model capabilities. Our goal is to show that cycle consistency is an emergent property in image-to-text and text-to-image models, and strongly correlates with various desired properties such as descriptiveness and reduced hallucination in text, and enhanced prompt-following in images (Section 4). Based on our findings, along with evidence of existing high-performing models which **already** incorporate cycle consistency in training [6-9], we believe that enforcing cycle consistency can be a helpful training objective for future research. For instance, text-to-image models such as DALLE-3 [6] and SD3 [7] report training on descriptive captions generated by an image captioning model. This process can be formalized as $\text{argmin}_G \ L(I, G(F(I)))$ where $I$ is the input image, $F$ is the image-to-text model and $G$ is the text-to-image model. Training a text-to-image model on synthetic captions is equivalent to enforcing image cycle consistency relative to the fixed image captioner. For vision-language models (VLMs), Synth2 [8] trains a VLM using data from a pretrained text-to-image model, while ITIT [9] jointly trains image-to-text and text-to-image models to be cycle-consistent. By injecting cycle consistency during training, both Synth2 and ITIT achieve high performance with significantly fewer data examples than state-of-the-art models.

---

> ### Author Response · Authors · 2024-11-28
> **Individual response to 31TL (2)**
>
> ### Q3. The claim that "more descriptive text leads to a broader distribution of generated images" is not convincing.
> We agree that the existing experiment does not properly control caption length – instead it varies the number of elements described. To address this issue, we make the following changes:
> 1. We replace Section 4 with a new experiment that studies **cycle consistency as a function of caption density** (see updated Section 4.4).  We better control the amount of information by summarizing captions from the DCI dataset [10] into varying lengths (5, 10, 20, 30, and 50 words) using LLaMA3-8B-Instruct. We find that cycle consistency improves as captions become more descriptive and dense, especially for the higher performing models FLUX-Time and SD3.
>
> 2. Congruent with the tags experiment, we report **diversity as a function of caption density**, measured by DreamSim pairwise distance between generated images using 10 different random seeds for the same caption. However, we observe inconsistent trends across the text-to-image models (see updated Figure 19). This discrepancy likely stems from differences in the experiment design: previous Section 4 used hierarchically created captions with tags, often altering meaning by introducing new elements, while the caption density experiment used summarized versions of the same caption with varying levels of detail. By focusing on caption density rather than tags, we aim to better understand how caption descriptiveness influences cycle consistency.
>
>
> ### Q4. Unaddressed claims in the abstract.
> Thank you for pointing these claims out and we intend to clarify them both here and in the updated abstract:
> 1. **Analyze failure cases**: We provide examples of failure cases of cycle consistency in Figures 23 and 24.  Failures include: synthetic images with artifacts or implausible generations but little effect on captions, descriptions of non-existent objects, endpoint model failures (i.e., the intermediate image or text representation is reasonable but the endpoint model creates inaccuracies which affect reconstruction). Many of these mistakes can be attributed to model error and usually affect text cycle consistency much more than image, mainly because images generated from incorrect captions often have lower cycle consistency, whereas image-to-text models do not always notice inaccuracies in synthetic images.
> There are also examples, typically for image cycle consistency, where information is not explicitly conveyed by the intermediate text, but the image reconstruction is nearly successful. To explain this, we can attribute cycle consistency as both a function of the intermediate representation **and the input**. For example, for image cycle consistency we find it is easy to reconstruct common or cliche images in very short captions. We show examples in Figure 9.
> 2. **How descriptiveness affects achieving cycle consistency**: We modify Section 4.4 to study caption density and cycle consistency, and find that more descriptive text positively influences cycle consistency as detailed in Q3.
> 3. **There are no explorations of training cycle-consistent models in the paper**: Our intent for Section 5 was to study variance of cycle consistency, as both image-to-text and text-to-image mappings can be stochastic due to random seed choice, temperature sampling, and prompt wording. We apologize for causing confusion and we have updated the abstract for clarity.
>
> ### Q5. Is image-text cycle consistency a meaningful metric for model development?
> Yes, we believe that image-text cycle consistency is a meaningful metric for model development. In the update paper, we show results indicating that more informative and descriptive captions correlate with cycle consistency, and similarly for generated images which are informative and faithful to their input text prompts. Qualitatively we highlight examples where better cycle consistency aligns with more preservation of information in Figures 2, 3, and 8. We also provide evidence of existing models that incorporate cycle consistency in training [6-9] in **Q2**.
>
> [1] Blip-2: Bootstrapping language-image pre-training with frozen image encoders and large language models. PMLR 2023.
>
> [2] Visual instruction tuning. NeurIPS 2023.
>
> [3] Improved baselines with visual instruction tuning. 2023.
>
> [4] Internvl-2.0. OpenGVLab, 2024. https://internvl.github.io/blog/2024-07-02-InternVL-2.0/
>
> [5] Llava-onevision: Easy visual task transfer. 2024.
>
> [6] Improving image generation with better captions. 2023.
>
> [7] Scaling rectified flow transformers for high-resolution image synthesis. ICML 2024.
>
> [8] Synth2: Boosting visual-language models with synthetic captions and image embeddings. Sharifzadeh et al., 2024.
>
> [9] Leveraging unpaired data for vision-language generative models via cycle consistency. Li et al., ICLR 2024.
>
> [10] A Picture is Worth More Than 77 Text Tokens: Evaluating CLIP-Style Models on Dense Captions, CVPR 2024.

---

> > ### Comment · Reviewer_31TL · 2024-12-03
> >
> > I appreciate the authors for their efforts in addressing the reviewers' feedback and making substantial changes to the manuscript.
> >
> > 1. However, I am concerned the extent of these modifications would be unfair as most experiments, analyses, figures, and tables have been replaced, making it a substantially different paper.
> >
> > 2. Despite these changes, I still believe the contribution of this paper is limited. The authors argue that ITIT explicitly trains cycle-consistent models while a few others implicitly encourage cycle consistency by training on synthetic data. While the paper highlights factors influencing cycle consistency between the trained T2I and I2T models, it does not provide evidence that improved cycle consistency translates to better model performance. This gap reduces the practical value of the findings for model development.

---

> > > ### Author Response · Authors · 2024-12-04
> > >
> > > ### Q1. Change in manuscript.
> > > We have mainly updated the manuscript to better communicate the results **requested by the reviewers**. However, we emphasize that the core topic, main results, and conclusions of the paper remain unchanged:
> > > * Figures 6 and 7: Trends remain the same, but we average across **all model combinations** and plot against both cycles to address concerns from Reviewers QcCd and BHgf.
> > > * Table 2: We plot against **cycle consistency** rather than diversity to address concerns from Reviewers QcCd and oP89.
> > > * Figure 12: Addresses concerns from Reviewers 31TL and oP89.
> > > * Table 1, Figure 4: Addresses concerns from Reviewers 31TL, go1x, and QcCd.
> > > * We also improved plot design and added qualitative visualizations to **enhance the quality** of the manuscript.
> > >
> > > We found the questions raised by the reviewers to be highly meaningful, accompanied by its results, which led to a **reorganization** of the sections. We are sincerely grateful for these insightful suggestions, which significantly enhanced the depth and quality of our analysis.
> > >
> > > ### Q2. Contributions.
> > > We clarify that the point of our paper is **not** to claim that training for cycle consistency translates to better model performance (although we will cite the prior work that shows this). Cycle consistency can offer practical value in the following ways: Given models that were **not explicitly trained to be cycle-consistent**, observing cycle consistency **at test time** strongly correlates with improved descriptiveness and reduced hallucination in generated text, and better prompt-following in images, i.e., several desired properties for a high-quality multimodal mapping. Therefore, it can be used as a **self-supervised** proxy for such performance measures.

---

### Official Review · Reviewer_QcCd · 2024-11-04

**Soundness:** 2
**Presentation:** 2
**Contribution:** 2
**Rating:** 5
**Confidence:** 3

**Summary:**

This paper analyzes the cycle consistency of current image/text generative models, which represents how well the original input is preserved when it consecutively passes through two generative models. To quantify the cycle consistency of images and text, the authors use metrics that measure perceptual similarity and present results for various combinations of image and text generative models. Using several benchmarks, the authors suggest that cycle consistency alone can imply the performance of models on downstream tasks by showing a high correlation between the two, thereby eliminating the need for creating additional test sets.

**Strengths:**

- The paper presents various quantitative analysis results on the proposed claim while also visualizing various qualitative results.
- The authors analyze a less explored aspect of generative models and provide insights into its significance.

**Weaknesses:**

- There is little analysis on the differences between the models used to measure cycle consistency. The paper simply mentions that recent models perform better, without analyzing whether the differences stem from their training data, objective functions, specific architectures, etc. Authors could have provided a table summarizing these differences and discussed how these factors may contribute to the observed performance differences in cycle consistency.
- In sections 4 and 5, it is unclear what message the authors are trying to convey. It is ambiguous how these sections relate to the cycle consistency discussed in sections 2 and 3. Authors could have better linked these sections to the overall narrative, such as explicitly stating how the divergence in text-to-image mappings (Section 4) and sensitivity in image-to-text mappings (Section 5) impact or relate to cycle consistency.

**Questions:**

- When calculating cycle consistency for each modality, one of two generative models is fixed. (SDXL Turbo for image cycle consistency / LLaVA 1.5-13b for text cycle consistency) Would results show the same trend if the fixed models were changed?
- If richer and more detailed data improves cycle consistency, can we say that recent models show better performance because they use quality data? It could lead to valuable insights if authors examined the training data characteristics of the better-performing models to see if there's a correlation with data quality, and discussed how this relates to cycle consistency performance.

---

> ### Author Response · Authors · 2024-11-28
> **Individual response to QcCd**
>
> Thank you for the insightful feedback and the helpful suggestions.
>
> ### Q1. What factors contribute to the observed differences in cycle consistency?
> As suggested by Reviewers go1x, QcCd, and 31TL, we have included an analysis of factors contributing to cycle consistency in the updated Section 3. The main findings are highlighted as follows:
> 1. **Cycle consistency improves with LLM scale**. An image-to-text model consists of a vision encoder, a projector, and a large language model (LLM). Scaling the vision transformer (ViT) for the vision encoder is reported to enhance performance [1], yet a simple MLP projection remains the dominant approach [2-5]. Since no model provides open-sourced weights with varying vision encoder scales, we focus our analysis on ablating the scale of the LLM. Figure 4 demonstrates that scaling the LLM enhances image and text cycle consistency across all image-to-text model families. Figure 5 visualizes this effect—despite being trained on the same dataset and architecture, only InternVL2-40B successfully captures both the color and the presence of a corner turret.
>
> 2. **Cycle consistency improves with re-captioned dataset quality**. Table 1 demonstrates that the quality of the re-captioned dataset (e.g., dataset re-captioned by GPT-4V, LLaVA1.6-34B) plays an important role in improving image cycle consistency, often outperforming models trained on larger datasets annotated by less-performant models (e.g., BLIP). On the other hand, text cycle consistency shows little difference between the LLaVA models, as the input text from sDCI often lacks fine-grained detail (evidenced in Figure 16) compared to longer and more descriptive synthetic captions, such as those produced by LLaVA1.6 and LLaVA-OV. We believe higher-quality human annotations and text-to-image models with longer context would enhance the analysis of text cycle consistency. We exclude InternVL2 from this analysis as its pre-training dataset details are not disclosed.
> We find minimal differences in objective functions across models: most image-to-text models use visual instruction tuning with auto-regressive objectives except BLIP2, and most text-to-image models are LDMs, except Stable Diffusion 3 with rectified flow. Therefore, we exclude analysis of the objective function. As suggested by the reviewer, we detail differences in architecture, scale, and dataset in Table 1, 5, and 6.
>
> ### Q2. It is unclear how Section 4 and 5 relate to cycle consistency.
> Previously Section 4 lacked proper control of caption length (mentioned by 31TL, op89) and Section 5 combined changes in length and style during caption rewriting, making it difficult to isolate their respective impacts on cycle consistency. To address these shortcomings and better relate Sections 4 and 5 to cycle consistency, we make the following changes:
> 1. Updated Section 4.4 studies **cycle consistency as a function of caption length**. Captions from the Densely Captioned Images dataset [6] are summarized into varying lengths (5, 10, 20, 30, and 50 words) using LLaMA3-8B-Instruct. Figure 12 shows that cycle consistency improves as captions become more descriptive, especially for the higher performing models FLUX-Time and SD3.
> 2. Updated Section 5 studies the **variance in cycle consistency** (formerly called sensitivity). Unlike the previous experiment, we address the effect of caption length separately in Section 4.4, and focus on sources of variance in this section. Specifically, we analyze how random seed selection, prompt style, and temperature sampling contribute to this variance. Table 2 shows that image-to-text models exhibit higher variance due to temperature sampling but remain relatively robust to changes in prompt style. In contrast, text-to-image models are significantly more sensitive to prompt style than to random seed sampling. Note that we excluded InternVL2-40B from measuring cycle consistency due to lack of compute, and we will add it to the final manuscript.
> 3. Congruent with the tags experiment, updated Figure19 shows **diversity as a function of caption density**, measured by DreamSim pairwise distance between generated images using 10 different random seeds for the same caption. However, we observe inconsistent trends across the text-to-image models. This discrepancy likely stems from differences in the experiment design: previously Section 4 used hierarchically created captions with tags, which altered meaning by introducing new elements. Instead, the caption density experiment uses summarized versions of the same caption with varying levels of detail. By focusing on caption density rather than tags, we aim to better understand how caption descriptiveness influences cycle consistency.

---

> ### Author Response · Authors · 2024-11-28
> **Individual response to QcCd (2)**
>
> ### Q3. Model ablation for measuring cycle consistency.
> As suggested, we update Figure 6, 7, and 10 to report cycle consistency **averaged across all models**, instead of just fixing one model in the pipeline. We also extend the analysis to **include all four combinations**, additionally comparing text quality (descriptiveness, hallucination) and image quality (prompt-following) with both image and text cycle consistency. We observe that both cycles exhibit a **strong correlation** across modalities, with text cycle consistency being more prominent.
>
> As requested, we report the Pearson correlation coefficient **per model**. The **correlation is consistently strong** for most models ($R^2 > 0.65$), except for BLIP2-2.7B and LLaVA-OV-0.5B with lower coefficients of 0.349 and 0.241, respectively. We attribute the low correlation to their use of small-scale, less-performant language models (OPT-2.7B, Qwen2-0.5B) as pre-trained backbones, which may cause poorer text reconstruction.
>
> | Fixed I2T Model | Text Cycle Consistency vs T2I Model Performance ($R^2$) |
> |-----------|-----------|
> | BLIP2-2.7B | 0.349 |
> | BLIP2-6.7B | 0.657 |
> | BLIP2-T5-xxl | 0.871 |
> | LLaVA1.5-7B | 0.966 |
> | LLaVA1.5-13B | 0.964 |
> | LLaVA-OV-0.5B | 0.201 |
> | LLaVA-OV-7B | 0.910 |
> | LLaVA1.6-7B | 0.963 |
> | LLaVA1.6-34B | 0.952 |
> | InternVL2-2B | 0.935 |
> | InternVL2-8B | 0.904 |
> | InternVL2-26B | 0.879 |
> | InternVL2-40B | 0.913 |
> | Average | 0.950 |
>
> |Fixed I2T Model| Image Cycle Consistency vs T2I Model Performance  ($R^2$) |
> |-----------|-----------|
> | BLIP2-2.7B | 0.836 |
> | BLIP2-6.7B | 0.834 |
> | BLIP2-T5-xxl | 0.879 |
> | LLaVA1.5-7B | 0.915 |
> | LLaVA1.5-13B | 0.916 |
> | LLaVA-OV-0.5B | 0.932 |
> | LLaVA-OV-7B | 0.954 |
> | LLaVA1.6-7B | 0.942 |
> | LLaVA1.6-34B | 0.953 |
> | InternVL2-2B | 0.904 |
> | InternVL2-8B | 0.903 |
> | InternVL2-26B | 0.880 |
> | InternVL2-40B | 0.902 |
> | All Models | 0.924 |
>
> |Fixed T2I Model| Text Cycle Consistency vs I2T Model Performance ($R^2$) |
> |-----------|-----------|
> | SD1.5 | 0.875 |
> | SDXL-Turbo | 0.845 |
> | SDXL | 0.879 |
> | SD3 | 0.870 |
> | FLUX Time | 0.861 |
> | All Models | 0.864 |
>
> |Fixed T2I Model| Image Cycle Consistency vs I2T Model Performance ($R^2$) |
> |-----------|-----------|
> | SD1.5 | 0.741 |
> | SDXL-Turbo | 0.759 |
> | SDXL | 0.731 |
> | SD3 | 0.790 |
> | FLUX Time | 0.794 |
> | All Models | 0.766 |
>
>
> ### Q4. Does cycle consistency correlate with data quality?
> Yes! Thanks to your valuable suggestions, we have discovered that the quality of the **re-captioned dataset** is associated withcycle consistency (mentioned in **Q1**). Table 1 details the re-captioned dataset for each model and their cycle consistency.
>
>
> [1] Blip-2: Bootstrapping language-image pre-training with frozen image encoders and large language models. Li et al., PMLR 2023.
>
> [2] Visual instruction tuning. Liu et al., NeurIPS 2023.
>
> [3] Improved baselines with visual instruction tuning. Liu et al., 2023.
>
> [4] Internvl-2.0. OpenGVLab Team, 2024. https://internvl.github.io/blog/2024-07-02-InternVL-2.0/
>
> [5] Llava-onevision: Easy visual task transfer. Li et al., 2024.
>
> [6] A Picture is Worth More Than 77 Text Tokens: Evaluating CLIP-Style Models on Dense Captions, Urbanek et al., CVPR 2024.

---

### Official Review · Reviewer_go1x · 2024-11-04

**Soundness:** 2
**Presentation:** 3
**Contribution:** 2
**Rating:** 5
**Confidence:** 4

**Summary:**

The paper analyzes the cycle consistency of image-to-text and text-to-image models. The study shows that while current models exhibit a level of cycle consistency, there is room for improvement, especially T2I models are sensitive to slight changes in prompts.

**Strengths:**

1. The paper focuses on interesting topics about the cycle consistency of by analyzing the cycle consistency of T2I and I2T models.
2. It provides a comprehensive analysis of cycle consistency in image-to-text and text-to-image mappings, highlighting the correlation between cycle consistency and downstream performance in tasks such as image captioning and text-to-image generation.

**Weaknesses:**

1. Although the paper presents that T2I models are more sensitive to small changes in input prompts, it lacks an in-depth analysis of why different combinations of T2I and I2T models yield varying performance. For example, how does the training dataset affect the cycle consistency? How does the pre-trained model in T2I or I2T affect the cycle consistency?
2. The paper does not sufficiently analyze why specific combinations of I2T and T2I models perform differently in terms of image and text cycle consistency. For example, BLIP2 underperforms compared to LLaVA1.6 in image cycle consistency while surpassing it in text cycle consistency.
3. The analysis in the paper highlights that text-to-image models are highly sensitive to slight changes in prompt structure (word choice, order, and length), which can lead to inconsistencies. However, the paper stops short of proposing concrete solutions or mitigation strategies for this issue.
4. The evaluation conducted solely on 1k MS COCO data is limited, especially since MS COCO captions often lack detailed descriptions of the images

**Questions:**

1. Recent research shows the hallucination problems in Multimodal LLM and compositional problems in T2I. How can the proposed method avoid this issue? For example, an input prompt could result in the generation of an incorrect image, which might then lead to an MLLM producing captions that are incorrect but resemble the original prompt. In this case, the cycle consistency might be high, but the actual performance should be low.
2. What is the cycle consistency on long captions?

---

> ### Author Response · Authors · 2024-11-28
> **Individual response to go1x**
>
> Thank you for the insightful review and the helpful feedback.
>
>
> ### Q1. What factors affect cycle consistency?
> As suggested by Reviewers go1x, QcCd, and 31TL, we have included an analysis of factors contributing to cycle consistency in the updated Section 3. The main findings are highlighted as follows:
> 1. **Cycle consistency improves with LLM scale.** An image-to-text model consists of a vision encoder, a projector, and a large language model (LLM). Scaling the vision transformer (ViT) for the vision encoder is reported to enhance performance [1], yet a simple MLP projection remains the dominant approach [2-5]. Since no model provides open-sourced weights with varying vision encoder scales, we focus our analysis on ablating the scale of the LLM. Figure 4 demonstrates that scaling the LLM enhances image and text cycle consistency across all image-to-text model families. Figure 5 visualizes this effect—despite being trained on the same dataset and architecture, only InternVL2-40B successfully captures both the color and the presence of a corner turret.
> 2. **Cycle consistency improves with re-captioned dataset quality**. Table 1 demonstrates that the quality of the re-captioned dataset (e.g., dataset re-captioned by GPT-4V, LLaVA1.6-34B) plays an important role in improving image cycle consistency, often outperforming models trained on larger datasets annotated by less-performant models (e.g., BLIP). On the other hand, text cycle consistency shows little difference between the LLaVA models, as the input text from sDCI often lacks fine-grained detail (evidenced in Figure 16) compared to longer and more descriptive synthetic captions, such as those produced by LLaVA1.6 and LLaVA-OV. We believe higher-quality human annotations and text-to-image models with longer context would enhance the analysis of text cycle consistency. We exclude InternVL2 from this analysis as its pre-training dataset details are not disclosed.
> We detail differences in architecture, scale, and dataset in Tables 1, 5, 6.
>
>
> ### Q2. Why specific combinations of I2T and T2I models perform differently in image and text cycle consistency.
> Most of the model combinations perform similarly on our updated results using the Densely Captioned Images dataset. However, there are exceptions: LLaVA1.5-13B achieves higher **text** cycle consistency than image cycle consistency relative to the other VLMs, whereas LLaVA-OV-7B has a higher consistency on **image** over text, as shown in Figure 14,15 (Appendix). We think this may be due to several reasons: LLaVA1.5 captions tend to point out less specific details, whereas other models try to pinpoint exact locations or meanings of photographs. Of course this is desirable for describing real images, but leads to variability when reconstructing text from synthetic images.. Secondly, LLaVA1.5 uses Vicuna-1.5 as its LLM which is a finetuned version of Llama 2. The sDCI captions which we use as our text inputs, are summaries of the full captions from DCI created by Llama2. Because the input text and output texts are created by similar models, it is likely that this contributes to their high alignment scores.
>
>
> ### Q3. The paper does not propose solutions for prompt sensitivity.
> We clarify that we are **not proposing a new method**, , but aim to provide an **empirical study on cycle consistency in image-text mappings**.
>
> As suggested by Reviewers QcCd and uoP89, we extend our analysis to study how **random seed selection, prompt and caption style, and temperature sampling** contribute to variance in cycle consistency (updated Section 5). Table 2 shows that image-to-text models exhibit higher variance due to temperature sampling but remain relatively robust to changes in prompt style. In contrast, text-to-image models are significantly more sensitive to caption style than to random seed sampling. Note that we excluded InternVL2-40B due to lack of compute, and we will add it to the final manuscript.
>
>
> ### Q4. MS COCO captions often lack detailed descriptions of the images.
> We agree with the reviewer’s suggestion and have replaced MS COCO with Densely Captioned Images (DCI) dataset [6], which features **higher-resolution** images annotated with more **dense captions**. Due to the limited prompt length of text-to-image models, we use sDCI, i.e., LLM-summarized DCI captions to fit 77 tokens, and sample 1K from the train split. Average image resolution and number of CLIP tokens per caption are as follows:
>
> | Dataset | Resolution | Tokens/Cap |
> |-----------|-----------|-----------|
> | sDCI | 1500×2250 pixels | 49.21 |
> | COCO | 480×640 pixels | 13.54 |
>
> Improving dataset quality revealed key factors influencing cycle consistency (updated Section 3.2). We thank the reviewer for the insightful suggestion.

---

> > ### Author Response · Authors · 2024-11-28
> > **Individual response to go1x (2)**
> >
> > ### Q5. How can the proposed method avoid hallucination in I2T and compositionality problem in T2I?
> > We clarify that we are **not proposing a new method**, but aim to provide an **empirical study on cycle consistency in image-text mappings**. We have updated Section 4.3 to study the relationship between object hallucination in text and cycle consistency. Contrary to the reviewer’s concern, we observe that cycle consistency strongly correlates with **reduced hallucination** in text (updated Figure 10 and 11, Section 4.3) and **improved compositionality** in images (Figure 6, Section 4.2).
> >
> > Note that in Section 4, caption quality generated by LLaVA1.6 and LLaVA-OV is lower compared to results in other sections, due to using suboptimal prompts. We will update the results in the final manuscript following implementation details in Appendix A1.
> >
> >
> > ### Q6. Cycle consistency on long captions.
> > As suggested, we have added Section 4.4 analyzing the effect of caption length on cycle consistency. We summarize captions from the DCI dataset into varying lengths (5, 10, 20, 30, and 50 words) using LLaMA3-8B-Instruct. We stop at 50 tokens to not overflow the token limit for text-to-image models. Updated Figure 12 shows that cycle consistency improves as captions become more descriptive and dense, especially for the higher performing models FLUX-Time and SD3.
> >
> > [1] Blip-2: Bootstrapping language-image pre-training with frozen image encoders and large language models. Li et al., PMLR 2023.
> >
> > [2] Visual instruction tuning. Liu et al., NeurIPS 2023.
> >
> > [3] Improved baselines with visual instruction tuning. Liu et al., 2023.
> >
> > [4] Internvl-2.0. OpenGVLab Team, 2024. https://internvl.github.io/blog/2024-07-02-InternVL-2.0/.
> >
> > [5] Llava-onevision: Easy visual task transfer. Li et al., 2024.
> >
> > [6] A Picture is Worth More Than 77 Text Tokens: Evaluating CLIP-Style Models on Dense Captions, Urbanek et al., CVPR 2024.

---

### Author Response · Authors · 2024-11-28
**Common Response to Reviewers**

We thank the reviewers for their helpful feedback and thoughtful insights. We are pleased that the reviewers found our topic **interesting** (go1x, oP89, QcCd), our analysis **comprehensive and thorough** (go1x, 31TL, BHgf), and **well presented** (31TL, oP89).

Firstly, we would like to clarify the goal of our paper is to present an **empirical study of cycle consistency in image-text mappings**. We observe growing cycle consistency across a wide range of image-to-text and text-to-image models, i.e., images and text are becoming increasingly interchangeable in their representations. We analyze 1) what factors are driving this trend, 2) what kinds of images and texts are exchangeable (i.e., cycle-consistent), 3) and sources of variance in cycle consistency.

We summarize our main updates to the paper as follows:

### Paper Reorganization:
**Section 3: Analysis on factors contributing to cycle consistency**.
* As suggested by go1x, QcCd, 31TL, we have included an analysis of factors contributing to cycle consistency.
* We find increasing cycle consistency with LLM scale and with high-quality dataset re-captioning.

**Section 4: Properties of cycle-consistent texts and images.**
* As mentioned by go1x, we include analysis on object hallucination in text (Section 4.3).
* As suggested by oP89, 31TL, QcCd, we study cycle consistency as a function of caption length (Section 4.4).

**Section 5: Variance in cycle consistency** (formerly called divergence/sensitivity).
* As suggested by QcCD, oP89, we relate Sections 4 and 5 to cycle consistency by analyzing how random seed selection, temperature sampling, and prompt and caption style cause variance in cycle consistency.
* Unlike the previous experiment, we address the effect of caption length separately in Section 4.4, and focus on sources of variance in this section.

### Dataset Updates:
* As suggested by go1x, we substitute COCO with the **Densely Captioned Images** (DCI) [1] dataset and update cycle consistency results accordingly. The DCI dataset features **higher resolution images** annotated with more **detailed captions** compared to COCO, improving the quality of our analysis.

### Model Updates:
* To analyze factors driving cycle consistency, we select image-to-text models with **disclosed** architecture, scale, and dataset details and weights.

[1] A Picture is Worth More Than 77 Text Tokens: Evaluating CLIP-Style Models on Dense Captions, Urbanek et al., CVPR 2024.

---

### Meta-Review · Area_Chair_nf9S · 2024-12-18

**Metareview:**

This paper received ratings of 5, 5, 5, 3, 3, and was unanimously recommended for rejection by all reviewers.

The paper study the cycle consistency of image-text mappings using combinations of text-to-image (T2I) and image-to-text (I2T) models. The authors observe a positive correlation between cycle consistency and certain model attributes such as descriptiveness, prompt-following ability, and reduced hallucination. They further evaluate factors affecting cycle consistency, including language model scale, dataset quality, and prompt style.

While this work provides an empirical exploration of cycle consistency in multimodal mappings, the primary contribution is based on observations rather than novel methods. It analyzes the correlation between cycle consistency and downstream task performance but does not focus much on methodologies or conclusive insights for improving model performance beyond existing literature.

Strengths:
- This study tackles an less-explored area, offering a detailed empirical analysis of cycle consistency in multimodal systems.
- The analysis spans multiple off-the-shelf models and considers factors like language model scale, dataset quality, and variance in prompts.
- The experiments are clearly presented with adequate visualizations and metrics.

Area for improvements:
- Limited novelty - the paper primarily presents observations without introducing a novel methodology or theoretical contribution. Many conclusions reiterate well-understood dependencies (e.g., high-quality datasets improve model performance).
- Although cycle consistency is analyzed, the paper lacks clear evidence or arguments for its practical significance in real-world applications.
- Major revisions during the rebuttal phase introduced significant changes to the experiments, conclusions, and structure of the paper. This makes it challenging to assess the original contribution versus the revised content.
- Insufficient depth in causal analysis. Thsi paper does not convincingly establish whether cycle consistency is a cause of improved performance or merely a byproduct of better models.

Despite its merits, the submission falls short in terms of methodological novelty, practical utility, and conclusive contributions to the field. It does not meet the bar for acceptance at ICLR. We encourage the authors to address the feedback provided in the reviews and consider submitting a revised version in the future.

**Additional Comments On Reviewer Discussion:**

The authors provided extensive revisions, including new experiments analyzing caption density, variance sources in cycle consistency, and better aligning the findings with cycle consistency metrics. The author further clarified that the paper is an empirical study, not proposing new methods, and positioned cycle consistency as a self-supervised heuristic for understanding model behavior and dataset quality. Reviewers appreciated the improved clarity and responsiveness but remained concerned about the substantial post-submission changes, which significantly altered the original paper.

While the revisions addressed many technical concerns, reviewers found the contributions incremental and largely observational, with limited practical utility or methodological novelty. The lack of causal evidence linking cycle consistency to downstream performance, coupled with the extensive restructuring during rebuttal. Therefore, a consensus has been reached that the submission does not meet the ICLR bar for acceptance.

---

### Decision · Program_Chairs · 2025-01-22

Reject